# *In situ* observations of turbulent ship wakes and their spatiotemporal extent

Amanda T. Nylund[1], Lars Arneborg[2], Anders Tengberg[1], Ulf Mallast[3], Ida-Maja Hassellöv[1]

[1]Department of Mechanics and Maritime Sciences, Chalmers University of Technology, Gothenburg, 412 96 Gothenburg, Sweden.
[2]Swedish Meteorological and Hydrological Institute (SMHI), Gothenburg, 426 71 Västra Frölunda, Sweden.
[3]Department Monitoring and Exploration Technologies, Helmholtz Centre for Environmental Research, Leipzig, 04318 Leipzig, Germany.

*Correspondence to*: Ida-Maja Hassellöv (ida-maja@chalmers.se)

**Abstract.** In areas of intensive ship traffic, ships pass every ten minutes. Considering the amount of ship traffic and the predicted increase in global maritime trade, there is a need to consider all type of impacts shipping has on the marine environment. While the awareness about, and efforts to reduce, chemical pollution from ships is increasing, less in known about physical disturbances and ship-induced turbulence has so far been completely neglected. To address the potential importance of ship-induced turbulence on e.g. gas exchange, dispersion of pollutants, and biogeochemical processes, a characterisation of the temporal and spatial scales of the turbulent wake is needed. Currently, field measurements of turbulent wakes of real-size ships are lacking, and this study addresses that gap by *in situ* and *ex situ* measurements of the depth, width, length, intensity and longevity of the turbulent wake for ~240 ship passages of differently sized ships. A bottom-mounted Acoustic Doppler Current Profiler (ADCP) was placed at 32 m depth below the ship lane outside Gothenburg harbour, and used to measure turbulent wake depth and temporal longevity. In addition, water temperature was used as a proxy for the water mass effected by the turbulent wake (thermal wake), as lowered temperature in the ship wake indicates vertical mixing in a thermally stratified water column. Thermal satellite images of the Thermal Infrared Sensor (TIRS) onboard Landsat 8 were used to measure thermal wake width and spatial longevity, using satellite scenes from the major ship lane North of Bornholm, Baltic Sea. Automatic Information System (AIS) records from both the investigated areas were used to identify the ships inducing the wakes. The results from the ADCP measurements show median wake depths of ~10 m, and several occasions of wakes reaching depths > 18 m, which is in the same depth range as the seasonal thermocline in the Baltic Sea. The temporal longevity of the wakes had a median of around 8 min and several passages of > 20 min. The satellite analysis showed a median thermal wake length of 13.7 km, and the longest wake extended over 60 km, which would correspond to a temporal longevity of 1 h 42 min (for a ship speed of 20 knots). The median thermal wake width was 157.5 m. The measurements of the spatial and temporal scales are in line with previous studies, but the maximum turbulent wake depth (30.5 m) is deeper than previously reported. The results from this study show that ship-induced turbulence occurs at temporal and spatial scales large enough to imply that this process should be considered when estimating environmental impacts from shipping in areas with intense ship traffic. The derived ship induced vertical mixing

along with its frequency calls for a better characterisation of spatial and temporal development, given the location of the seasonal thermal stratification of 10–20 m depth and thus the possible impact on local biogeochemical cycles, gas exchange and nutrient distribution.

## 1 Introduction

The shipping industry holds a key role in today's society, as 80–90 % of all global trade is transported via ship (Balcombe et al., 2019). In areas of intensive ship traffic, e.g. in the Baltic Sea, there can be more than 50.000 ship passages annually, which in turn is approximately one ship passage every ten minutes (HELCOM, 2010). Yet, maritime trade is predicted to increase by 3.4 % annually until 2024 (UNCTAD, 2019). Transport by ship is also advocated as the most energy efficient as it in general has low carbon footprint per tonne and distance of transported goods (Balcombe et al., 2019). However, the carbon footprint is only one of many environmental impacts from shipping, and to fully estimate the impact of this growing industry, a holistic assessment is needed (Moldanová et al., 2018). To make a reliable holistic assessment, all types of impacts on the marine environment need to be considered, both from polluting and physical disturbances. This paper will focus on a previously disregarded physical disturbance from shipping, namely ship-induced turbulent wakes and their spatiotemporal extent.

When a ship moves through water, the hull and propeller create turbulence, which forms a turbulent wake behind the ship, characterised by an increased turbulence and a dense bubble cloud (NDRC, 1946; Soloviev et al., 2010; Voropayev et al., 2012; Francisco et al., 2017). Knowing and being able to properly characterise temporal and spatial scales of the turbulent wake has several reasons. It can be used to estimate the distribution of contaminants and pollutants discharged from ships (Katz et al., 2003; Loehr et al., 2006; Golbraikh and Beegle-Krause, 2020). Furthermore, the bubbles created in the turbulent wake can affect the gas exchange between ocean and atmosphere, in addition to the increased gas exchange due to the turbulence itself (Trevorrow et al., 1994; Weber et al., 2005; Emerson and Bushinsky, 2016). The episodic nature, intensity, and duration of the ship-induced turbulence is also of a magnitude that have been shown to affect the mortality of copepods and diatoms (Bickel et al., 2011; Garrison and Tang, 2014). Moreover, in areas with intense ship traffic, the ship-induced vertical mixing could possibly affect nutrient availability and natural biogeochemical cycles in seasonally stratified waters.

Up to now, the environmental impact of ship-induced vertical mixing has been overlooked, and there is a limited amount of field observations reporting spatiotemporal scales of the turbulent wake. There are few studies about ship-induced turbulence in general and none investigating the possible environmental impact of ship-induced vertical mixing. Remote sensing approaches focused on detecting wakes from a surveillance perspective (Fujimura et al., 2016) or the theoretical possibility of doing so (Issa and Daya, 2014). These approaches mainly rely on Synthetic Aperture Radar (SAR) to identify sea surface roughness. Other studies focused on the vertical distribution of the turbulent wake for military purposes, with the interest of

detecting the wake and minimizing the wake signal (Smirnov et al., 2005; Liefvendahl and Wikström, 2016). Moreover, the formation and distribution of the bubble cloud in the turbulent wake has been in focus, rather than the turbulence and mixing. Besides the different foci, most of the available studies are numerical modelling studies of ship wakes. Measurements are on model-scale ships for validation (Carrica et al., 1999; Parmhed and Svennberg, 2006; Fu and Wan, 2011; Liefvendahl and Wikström, 2016), which generally only resolve the wake for distances up to a ship length after the ship. In real world,

temporal and spatial scales of the turbulent wakes are significantly larger. Turbulent processes are difficult to investigate at laboratory scale, since the Reynolds number is much too small in the laboratory and the results can therefore not be expected to represent turbulence in nature.

The few studies that are based on field measurements or focus on the spatial and temporal scales of the turbulent wake,

report measured wake depths between 6–12 m (Table 1). Measured wake widths are more varied, with a range of 10–250 m (Table 1). This large variation could partly be due to the different methods used to define the wake region, as well as the difference in size and type of the investigated vessel. The longevity of the wake has been measured both as a temporal duration and as a length. Already in 1946, the United States National Defense Research Committee (US NDRC) reported detectable bubbles and temperature differences in the turbulent wake 30–60 min after ship passage. Trevorrow et al. (1994)

made measurements of the temporal scale of the turbulent wake and reported strong acoustic scatters from the bubbles in the wake for 7.5 min after passage. Soloviev et al. (2010) evenreported that bubbles from the turbulent wake were visible from 10–30 min after ship passage, corresponding to a distance of 4–10 km, for a ship with a speed of 12 knots. It becomes clear that the turbulent wake can reach depths of 10–15 m and can have a longevity of up to 30 min and/or 10 km. However, except Trevorrow et al. (1994) and NDRC (1946), information of wake width, length, or duration were always a by-product

of these studies. Therefore, they naturally lack simultaneous measurements of depth, width, and length of the turbulent wake, as well as a statistical sound and reliable data basis with higher number and variety of vessels (type, speed, size). Thus, there is currently too few field measurements of the turbulent wake of real-size ships, to reliably estimate the temporospatial scales of turbulent wakes (Carrica et al., 1999; Parmhed and Svennberg, 2006; Ermakov and Kapustin, 2010).

The aim of this study is therefore to obtain a reliable overview of the magnitude of the spatiotemporal influence of ship-induced vertical mixing through the integration of ~240 observations of ship passages, and a combination of methods to describe the depth, width, length, and longevity of the turbulent wake. As the study has been conducted *in situ* and *ex-situ*, on different temporal and spatial scales, and includes ships of different types and varying size, it constitutes a solid base for a first estimate of the order of magnitude of the spatiotemporal extent of turbulent ship wakes. A better understanding of the

spatial and temporal extent of the turbulent wake, is needed to identify where ship-induced vertical mixing could have a significant impact on local biogeochemical cycles, and thus should be studied further. Moreover, it provides a basis for estimating the summed wake area in a region, where an effect on gas exchange could be expected. In addition, it will provide valuable information for monitoring in areas with intense ship traffic, as well as for studies of the dispersion of pollutants

from ships. It will be of particular importance for the FerryBox community, as FerryBoxes perform continuous
measurements onboard ships en route, often in major ship lanes that may lead to biased results compared to surrounding
water. In short, increased knowledge about the spatiotemporal extent of turbulent ship wakes, makes it possible to identify
when and where ship-induced turbulence needs to be considered.

**Table 1. Previously reported field measurements of the spatial and temporal scales of the turbulent wake. The method used to**
**estimate the turbulent wake is indicated in the "Method" column. For studies where only the temporal wake longevity was**
**measured, an estimate of the wake length has been calculated using the wake duration and a ship speed of 12 knots.**

| Study | Method | Wake depth [m] | Wake length [km] | Wake duration [min] | Wake width [m] |
|---|---|---|---|---|---|
| NDRC (1946) | Acoustic/thermal | 3—10 | 11—22 | 30—60 | 40—90 |
| Trevorrow et al. (1994) | Acoustic | 6—12 | 2.8* | 7.5 | 66 (average) |
| Katz et al. (2003) | Dye concentration | 8—10 | 3** | | |
| Weber et al. (2005) | Acoustic | 8 | 6 | 15 | |
| Stanic et al. (2009) | Acoustic | | 1.5—2 | 20 | 10 |
| Ermakov & Kapustin (2010) | Acoustic | 4—8 | 3.7—5.5* | 10—15 | 40—80 |
| Soloviev et al. (2010) | Acoustic | 10—15 | 4—10* | 10—30 | |
| Gilman et al. (2011) | Visible surface trace | | | | 100—250 |
| Soloviev et al. (2012) | Acoustic | 7 | | | |
| Francisco et al. (2017) | Acoustic | 6—12 | 0.5* | 1.5 | |

*Calculated based on temporal longevity and a ship speed of 12 knots, **Distance at which the max width was documented

## 2 Materials and methods

The data collection was conducted in two parts: one field study in the large ship lane outside Gothenburg harbour, and a
satellite image analysis of sea surface temperature in the large ship lane north of Bornholm, Baltic Sea (Fig. 1). The field
study covered the vertical scale and the temporal longevity of the turbulent wake, and the satellite image analysis was used
to estimate the thermal wake width and spatial longevity.

### 2.1 Gothenburg harbour study

The field study was conducted off the Swedish west coast, in the large ship lane outside Gothenburg harbour (Fig. 1).
Gothenburg harbour is the largest harbour in Scandinavia, with 120 port calls per week, including large container ships, oil
tankers, car carriers, and passenger ferries (The Port of Gothenburg, 2020). The size of the harbour, the frequency of port
calls, and the variety of ship types, makes it a suitable study area for ship-induced vertical mixing. The site of instrument
deployment was outside the port area, under the fairway where all incoming large ships need to pass (Swedish Maritime
Administration, 2020). It was also inside the area where tugboats and pilots are required when applicable, but outside the
speed restriction area, thus ships were traveling at normal speed. For the *in situ* measurements, the Gothenburg site was

considered more suitable compared to the Bornholm study area, as it was more easily accessible and the risk of losing the instrument to other maritime activities was lower. The water depth at the study site was 32 m, which is similar to the water depth where the major ship lanes on the Swedish West and South coast are located (< 20 m and < 50 m respectively) (Jakobsson et al., 2019). In the Baltic Proper (Western and Eastern Gotland Basins, Northern Baltic Proper), the median depth is deeper (< 75 m), but the major ship lane pass south of Gotland, which is the shallowest part of the Baltic Proper
(approximately 25–30 m) (Jakobsson et al., 2019).

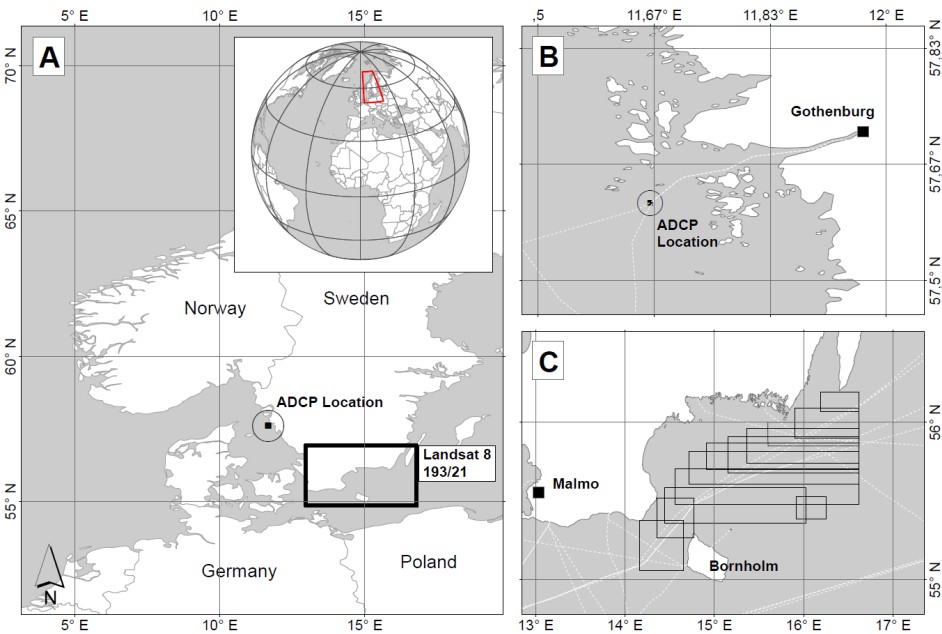

**Figure 1: Overview of the two study areas (a) showing the location of the ADCP under the ship lane outside Gothenburg (b) and the area covered by the analysed satellite images (c). White dashed lines indicate ship routes of ferry lines and the boxes in (c)**
**indicate the area defined as the ship lane area in the satellite image analysis. The ship lane and traffic separated zone north of Bornholm are shown in Figure 12.**

### 2.1.1 Field measurements and data collection

A bottom-mounted Nortek Signature 500 kHz broadband Acoustic Doppler Current Profiler (ADCP) was deployed under the ship lane (57.61178 N, 11.66102 E), fixed in upward-looking position in a bottom frame (Figure **2**). Similar setups have
previously been used to study the bubble cloud of the turbulent wake by Trevorrow et al. (1994) and Weber et al. (2005). The instrumental setup provides measurements of the overlaying water column trough time (Figure 3), hence, recording the wake development in a fixed point through time. Under the assumption of a stationary wake moving with the ship velocity, the observations can also be interpreted in terms of the spatial change of the wake with distance from the ship. The instrument was deployed at approximately 30 m depth, for a duration of 4 weeks (28 August to 25 September 2018). The
ADCP measured along beam current velocities, using four slanted beams (25° angle) and one vertical beam (ping frequency

1 Hz, cell size 1 m on all beams). The echo amplitudes from the beams were also used to detect the wake bubbles. All single ping data on currents and echo amplitude was stored on-board the instruments and analysed, see sect. 2.1.2. The range of sonar frequencies that are suitable for detecting bubbles in the turbulent ship wake is 30 kHz to 1 MHz and depends on the size of the bubbles in the wake (Liefvendahl and Wikström, 2016). A SonTek CastAway®-CTD (Xylem, San Diego,

California) was used to measure salinity and temperature profiles at the time of the instrument deployment (August 28, 2018, 4 casts) and retrieval (September 25, 2018, 4 casts).

A dataset of the ships passing the study area during the field measurement period was purchased from the Swedish Maritime Administration. The dataset is from the Baltic Marine Environment Protection Commission (HELCOM) Automatic

Information System (AIS) database, which is processed according to the procedure described in the annex of the HELCOM Assessment on maritime activities in the Baltic Sea 2018 (HELCOM, 2018). The Swedish Institute for the Marine Environment (SIME) provided additional files from the same HELCOM database, with AIS data for the analysed satellite scenes and the Gothenburg harbour study area. Vessel information from MarineTraffic – Global Ship Tracking Intelligence ([www.marinetraffic.com](www.marinetraffic.com)) was used to retrieve detailed information about the width, length and draught of the ships in the

dataset.

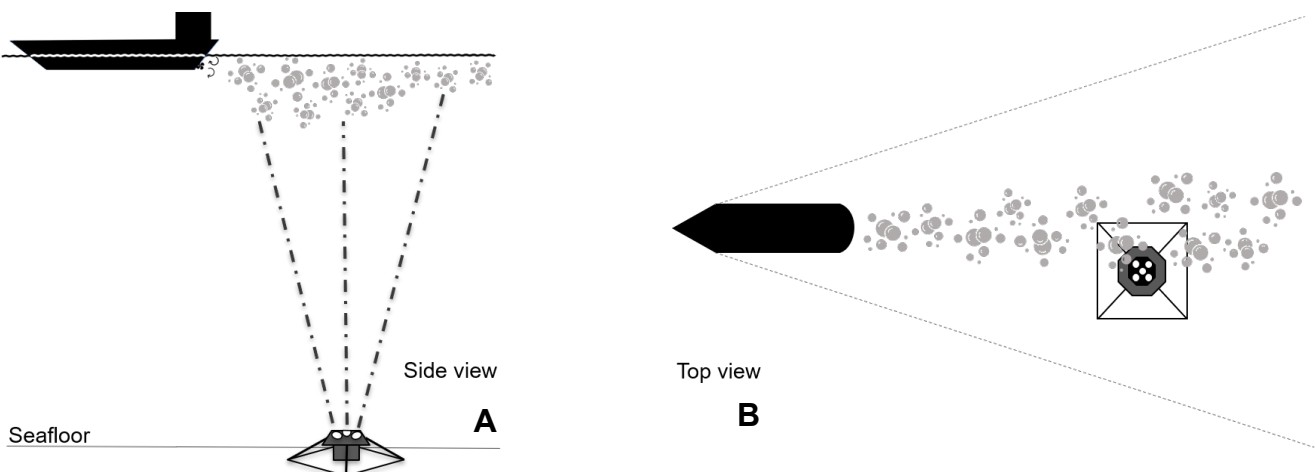

**Figure 2: Scheme of instrument deployment, showing a) the sideview with perspective of the ADCP placed on the seafloor facing upward and recording the turbulent wakes during ship passages, and b) a top view perspective of the ADCP recording bubbles from a turbulent wake, induced by a ship passing above, but slightly to the side of the instrument.**

### 2.1.2 Data analysis

**Compiling the ADCP wake dataset**

All ship wakes in the dataset were identified manually using high resolution figures of the echo amplitude of the ADCP beams (see Fig. 3 for example). As the bubbles in the turbulent wake reflect the sound more efficiently than water, it results in an elevated echo amplitude in the turbulent wake region (NDRC, 1946; Marmorino and Trump, 1996; Trevorrow et al., 1994; Weber et al., 2005; Ermakov and Kapustin, 2010; Francisco et al., 2017). Generally, the wake signal could be clearly distinguished from bubbles induced by waves or signal noise from fish or zooplankton. However, ambiguous cases were noted, and the wake dataset was therefore divided into wake categories based on the quality of the wake signal. Each wake in the dataset was then linked to a ship passing in the vicinity of the ADCP, using the HELCOM AIS dataset and manual comparison. This introduced additional uncertainties, as not all wakes had clear match with a ship passage. After incorporating the matching uncertainties, the final wake categories used in the analysis were: "**wake**", only including clear wakes with one clear match or delayed match; "**double**", clear wakes where two or three ships passed the instrument at the same time; and "**no wake**", which included all passages within 184 m of the instrument that did not induce a visible wake, as well as all uncertain wakes and matches, which were mostly due to windy conditions which created noisy data. The distance at which a wake can be detected from a passing ship is affected by wake broadening, drifting, and ship width. In this study, the 184 m radius was chosen, as it was the furthest distance at which a clear wake and match was found in the dataset. There were two factors contributing to the existence of the "**double**" category. Firstly, the turbulent wakes in the dataset could be detected from ships passing at distances up to 184 m from the ADCP instrument. Hence, in cases when two ships passed at similar distances from the instrument at the time of a detected wake, it was not possible to distinguish which of the ships induced the wake. Secondly, large ships may require pilot assistance and/or tugboats to enter the harbour. In these cases, the ships pass right next to each other and it is not possible to assign the wake to a single ship. Lastly, some wakes and passages were removed from the analysis altogether. These included ships with missing information in the AIS data (size information) and small sailing vessels, as they due to their small size and engine power were not deemed relevant for the investigated process.

**Distance calculation, AIS and ADCP dataset**

The AIS dataset included position reports for each ship every 2–10 seconds, which were used to calculate the ship's track. The closest distance between the ship-track and the vertical beam of the ADCP instrument was then calculated, using a local planar coordinate system, with the instrument at origo. The coordinates for the closest point on the track was also calculated, using the Python GeoPy package function distance.distance, and the points just before and after the closest point on the track were then identified.

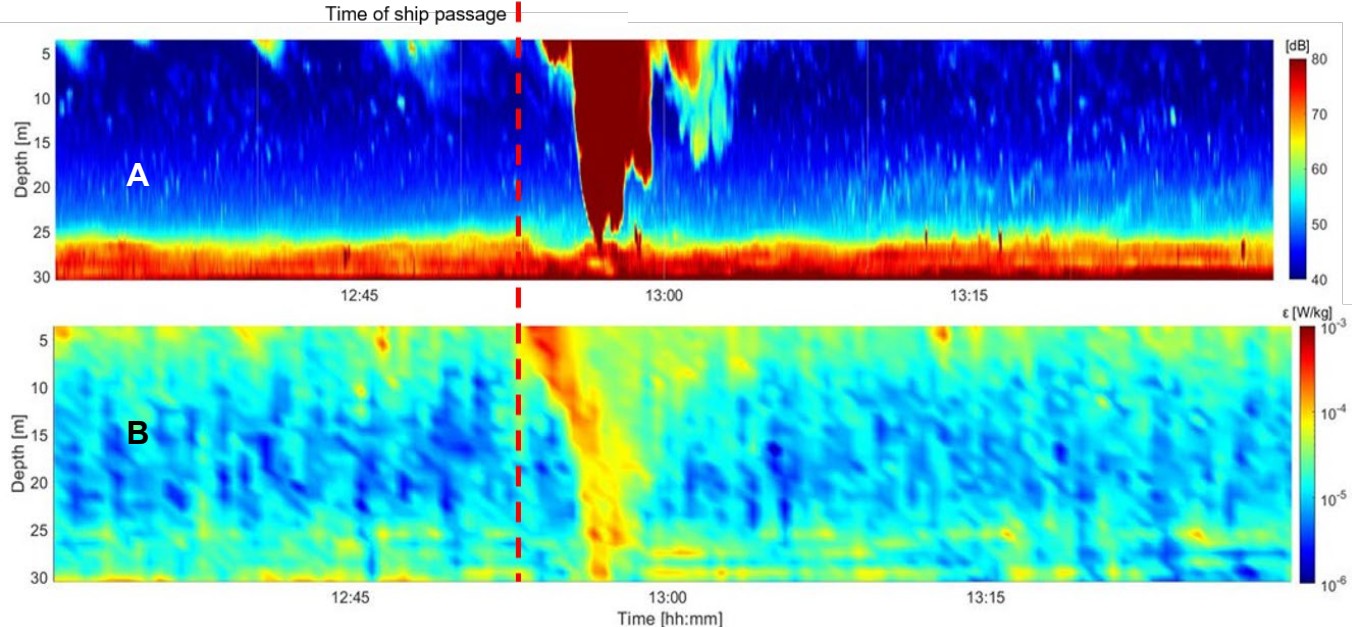

**Figure 3:** Example of the bubble wake signal in the echo amplitude dataset (a) and the calculated dissipation rate of turbulent kinetic energy, $\varepsilon$, (b) from one hour of ADCP measurements. The upward facing ADCP was placed at approximately 30 m depth, repeatedly measuring the water column in one point. The dashed red line marks the time of ship passage. The high intensity red (a) and yellow (b) areas after the ship passage represent the wake region. The increase of $\varepsilon$ down to the bottom is evidence of increased turbulence and a vertical mixing down to 30 m depth. The wake was induced by a cargo ship (width 25 m, length 229 m, draught 7 m), which passed the instrument at a distance of 34 m and a speed of 19 knots.

**Turbulence calculation, ADCP dataset**

The dissipation rate of turbulent kinetic energy ($\varepsilon$) is a measure for the strength of the turbulence. Per definition $\varepsilon$ is the rate of energy conversion from kinetic energy to heat due to viscous friction in the smallest eddies, but in a stratified water column $\varepsilon$ is also proportional to the mixing between different water masses. There are various ways of determining dissipation rates. In the present work $\varepsilon$ is estimated from the ADCP data using the structure function method (e.g. Lucas et al. (2014)), which estimates the dissipation rate of turbulent kinetic energy from the second-order structure function following Eq. (1):

$$D_{11}(r, \Delta r) = \overline{\left(u_r'(r) - u_r'(r + \Delta r)\right)^2}, \tag{1}$$

where $u_r'$ is the fluctuating velocity in the $r$-direction (in this case the beam direction), $\Delta r$ is the separation distance between two points along the beam, and overbar denotes time averaging. For separation distances shorter than the largest eddies the structure function relates to the dissipation rate and separation distance as in Eq. (2):

$$D_{11}(r, \Delta r) = C\varepsilon^{2/3}\Delta r^{2/3}, \tag{2}$$

where $C$ is a universal constant. Since the shortest distance (the ADCP bin size) was 1 m, the method is only expected to work for very strong turbulence with vertical eddy scales of magnitude larger than 2–3 m.

For each ship wake in the "**wake**" and "**double**" category, the along beam current velocity measurements from the ADCP were used for turbulence calculations in the wake region. One of the slanting beams was malfunctioning but the four remaining beams were analysed. A 1-hour dataset following each passage, identified by the start of the bubble cloud, was analysed. Spikes deviating more than four times the standard deviation from the mean in overlapping windows of 100 sec length were removed. Since the velocity signal of surface waves at different depths may be expected to be coherent whereas

turbulent signals are not, the two Empirical Orthogonal Function (EOF) modes with largest variance were removed from the series to reduce the influence of surface waves. A fourth order Butterworth high-pass filter with cut-off period 600 sec was used to extract the turbulent velocity fluctuations. The dissipation rate of turbulent kinetic energy was estimated in 30 sec bins using the structure function method according to the method described in Lucas et al. (2014). One dissipation rate estimate was based on the average of the result for the three slanting beams (see Fig. 2 for an example), and another was

based on the vertical beam.

**Calculating wake depth, longevity, and maximum $\varepsilon$ intensity, ADCP dataset**

    For each wake in the categories "**wake**" and "**double**", the wake region was defined for the parameters echo amplitude (bubble wake), dissipation rate of turbulent kinetic energy ($\varepsilon$), and the maximum velocity variance. To reduce noise in the

dataset induced by turbidity at the sea floor, the data was normalised with respect to vertical distance from the instrument, assuming exponential decay of the signal strength. The wake region was defined by visual scrutiny of echo amplitude and $\varepsilon$ figures (see Fig. 2 for an example) and manually annotated. The elevation in echo amplitude/$\varepsilon$ used for delimiting the wake region, as well as the depth and duration to consider, was manually adjusted for each wake to exclude noise. In general, the threshold was ~15% higher compared to the daily/nightly mean. The deepest part of the wake region was used as a measure

of the maximum wake depth and the maximum $\varepsilon$ intensity in the wake region was used as a measure of the maximum turbulence. The duration of the wake (temporal longevity in min) was calculated using the start time and end time of the wake region. All calculations were pursued using an individually developed Python code.

**Statistical analysis and graphical presentation of the ADCP wake dataset**

For the statistical analysis the categories "**wake**", "**double**", "**close wake**", and "**no wake**" were used. The category "**close wake**", comprises all wakes induced by ships passing within 0–3 ship widths from the instrument, which roughly corresponds to 75 m. This cut-off was chosen as there was a substantial decrease in the percentage of induced wakes at passages > 3 ship widths from the instrument, indicating difficulties in detecting wakes at larger distances. As the wakes in the "**double**" category lack information about the ship inducing the wake, no "**double**" wakes are included in the "**close**

**wake**" category. Hence the double category is presented separately. For each category the median wake depth (m) and temporal wake longevity (min), was calculated for the bubble wake and the $\varepsilon$ dissipation rate wake, together with standard

deviation (std) and the 25th and 75th percentile. Furthermore, the percentage of ship passages that induced a visible wake in the ADCP beams was calculated along with the maximum ε intensity in the wake region.

For the graphical presentation, the wake depth and longevity results are presented in relation to vessel force (F) [kg m s$^{-2}$]. F was calculated from the ship width (B) [m], draught (T) [m], and speed ($s$) [m s$^{-1}$], as in Eq. (3):

$$F = \rho * B * T * s^2 , \qquad\qquad\qquad (3)$$

with seawater density ($\rho$) equal to 1025 kg m$^{-3}$. The F parameter is proportional to ship drag and relates the wake depth and longevity to vessel size and speed, which are parameters affecting the formation of the turbulent wake.

**2.2 Bornholm satellite study**

The Bornholm study area was chosen, as it covers the most intensely trafficked ship lane in the Baltic Sea, with approximately 50,000 ship passages per year (HELCOM, 2010). All large ships heading for the Eastern and Northern ports of the Baltic Sea, must use the Bornholm ship lane (HELCOM, 2018), which makes it ideal for studying ship-induced vertical mixing from a variety of different ship types. Besides the purely traffic-related reason, a second reason for choosing
the Bornholm area compared to the Gothenburg area, in which in.-situ data (ADCP, CTD) was retrieved was the availability of cloud-free satellite scenes, which is essential for detecting any surface object in the optical and thermal wavelength. The Bornholm area (path 193/ row 21) had 23 scenes with less than 23% cloud cover above the sea until August 2018, for the Gothenburg area (path 196/ row 20) only 9 scenes were available.

**2.2.1 Data collection**

All required optical and thermal infrared data from Landsat 8 were retrieved from https://s3-us-west-2.amazonaws.com. The study area for the Bornholm area in the Baltic Sea was covered by path/row 193/21 (see Fig. 1 for overview of study area).

**2.2.2 Data analysis**

**Compiling the satellite dataset**
To obtain average wake lengths and widths indicating vertical mixing on regional scales, optical, near-infrared and thermal-
infrared bands from Landsat 8 were analysed. The dataset includes Landsat 8 data having a cloud cover < 23% (n=23). For optical and infrared data cloud coverage acts as opaque layer hindering to infer any information below it. The procedure includes a general and automatized data pre-processing scheme (Matlab), an automatic ship detection (Matlab) and a manual wake digitization (ArcMap). The pre-processing encompasses i) an automatic download of all available satellite scenes with less than 23% cloud coverage of the given path/row, ii) a masking of land areas using a combination of the modified
normalized difference water index (MNDWI) after Xu (2006) and a Otsu-based threshold procedure (Otsu, 1979), iii) a masking of opaque and cirrus clouds classified as such based on the CFMask (Foga et al., 2017), and iv) finally a conversion

from top-of-the-atmosphere (TOA) spectral radiances of band 10 to sea surface temperatures (SST) using transmission, downwelling and upwelling radiances modelled for each scene using a MODTRAN based online tool (Barsi et al., 2003).

Detecting ships was pursued semi-automatically following an optical approach similar to the one described by Heiselberg (2016). After masking, the remaining and analysable area is open water only. Spectrally, ships can be differentiated using the visual and short-wave-infrared part of the spectrum, even on the basis of coarser spatial resolution of 30 m as in the present case. As both parts of the spectrum are included in the MNDWI a global threshold of 0.09 was used on the MNDWI image for each scene to detect potential ships. To reduce the number of false positives due to unmasked cloud interference, a

further selection criterion was added, using optical ship wake characteristics described in Gilman et al. (2011) and Heiselberg (2016), which is also visible in MDWNI space. Around all potential ships, a search window of 15x15 pixel (450x450m) was created. If MNDWI values > 0.13 representing ship wakes was detected, the potential ship was converted to a true ship, while remaining potential ships were neglected.

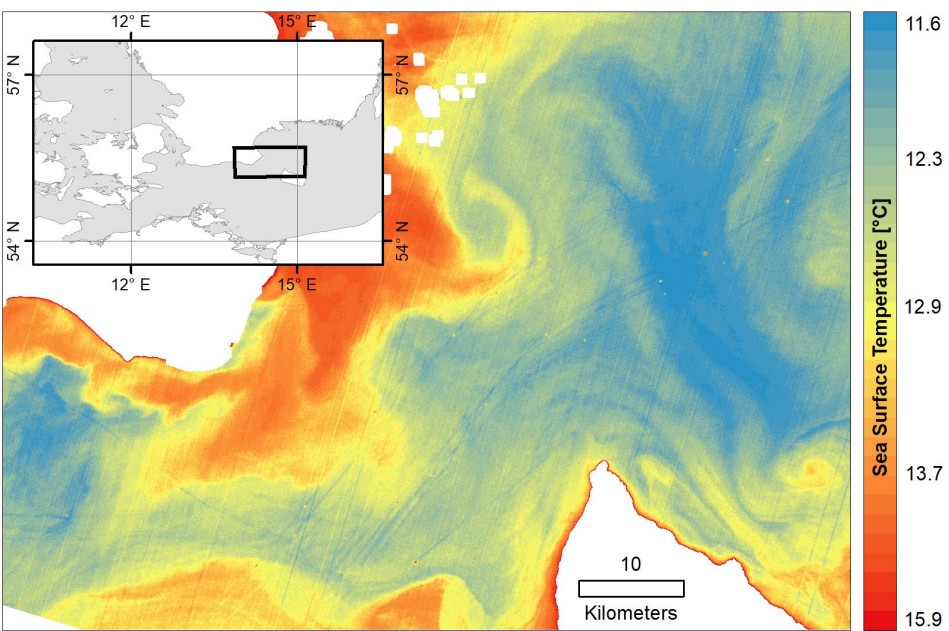

**Figure 4: Example of satellite scene with visible thermal wakes in the Bornholm study area. Note the ships visible as warmer yellow dots and the thermal wakes visible as colder blue lines stretching behind the ships. The oblique regular stripes are not ship wakes, but a sensor induced radiometric artefact. Landsat-8 image courtesy of the U.S. Geological Survey.**

Using the ships as spatial indication, all available 23 scenes were screened for thermally indicated ship wakes. In case of an occurrence, all thermal wakes for which a ship was detected, were digitalised. Using this approach, the wake lengths were

obtained (see Fig. 4 for example of visible thermal wakes). To also retrieve wake widths, cross profiles were subsequently created in intervals of 250 m along the thermal ship wake, with a length of 400 m each. The cross-profile lengths were

orientated at the maximum widths of <300 m presented in Gilman et al. (2011). Wake width was automatically determined analysing the local minima (thermal wake centre) and local maxima (surrounding uninfluenced water area) for each of the cross profiles.


**Combining the satellite wakes with AIS data**

Identified wakes and ships from satellite data were automatically matched against AIS data, to identify the ships inducing the wakes. All scenes were manually controlled to make sure the automatically matched ships were moving in the correct direction to have induced the wake. As the area of interest was the large ship lane north east of Bornholm, only the ships in

the traffic separated part of the ship lane stretching from Bornholm to Öland's south tip, were included in the analysis (see boxed area in Fig. 1c). In addition to the matched satellite ships, all other ships present in the area at the time of each satellite scene were identified.

**Statistical analysis of satellite wake dataset**

For the satellite dataset, the median spatial wake longevity (m) and wake width (m), was calculated, together with standard deviation (std) and the $25^{th}$ and $75^{th}$ percentile. The percentage of ship passages inducing visible thermal wakes, was also calculated.

**3. Results and discussion**

In the Gothenburg harbour study, there was a total of 96 detected turbulent wakes which could be successfully matched to a

passing ship. In the Bornholm satellite image analysis, 144 thermal wakes were detected in the ship lane area, and successfully matched to a ship. Thus, a total of 240 ship wakes were included in the analysis, and the results from each study area will be presented separately below.

**3.1 Gothenburg harbour study**

During the measurement period, there were a total of 413 ship passages within 184 m of the instrument. Of these passages,

there were 65 occasions when two ships passed the instrument at the same time. As a double ship passage only induces one wake, these occasions were considered as one passage when looking at the percentage of passages inducing wakes. In addition, 15 other passages were removed due to data uncertainties originating from entirely missing data (n=3), small size vessels irrelevant for the present study such as sailing/pleasure vessels (n=5), and multiple passages or wakes with unclear matches (n=7). This resulted in a total of 333 passages included in the analysis. 96 of those passages induced clearly visible

wakes (29 %) due to single ship passages (n=69) and double passages (n=27). The close wake passages (< 3 ship widths from the instrument) comprised 57 % (n=39) of all single ship passages. The statistical analysis of wake depth and longevity

is presented for the entire dataset in section 3.1.4 and 3.1.5, together with a graphical presentation of the close wake passage dataset.

### 3.1.1 Environmental parameters

At the time of deployment, there was a clear stratification at 10 m depth, with an upper mixed layer salinity of 25.5, and a gradual increase of salinity below the stratification, reaching a maximum salinity of 32 at 32 m depth (Fig. 4). The temperature profile showed a rather uniform profile, with only a slight increase towards the surface, indicating that salinity was the main stratifying component (Fig. 4). The surface layer had a temperature of 18–18.6 °C, the middle layer ranged from 17.6°C at 10 m to 17.3 °C at 20 m, and the deepest layer went from 17.4 °C to 16.4 at the sea floor. At the time of

instrument retrieval, there was only one clear pycnocline at 5 m depth, with an upper mixed layer temperature around 14 °C and salinity around 27. The temperature below the pycnocline was around 16 °C and the salinity was 33. This type of structure is usual in this area, as the Baltic Surface current which brings low saline water from the Baltic Sea is on top of the more saline water from the Skagerrak (Snoeijs-Leijonmalm and Andrén, 2017). Note that the water column is unstable in temperature, so also here salinity is the stratifying component.

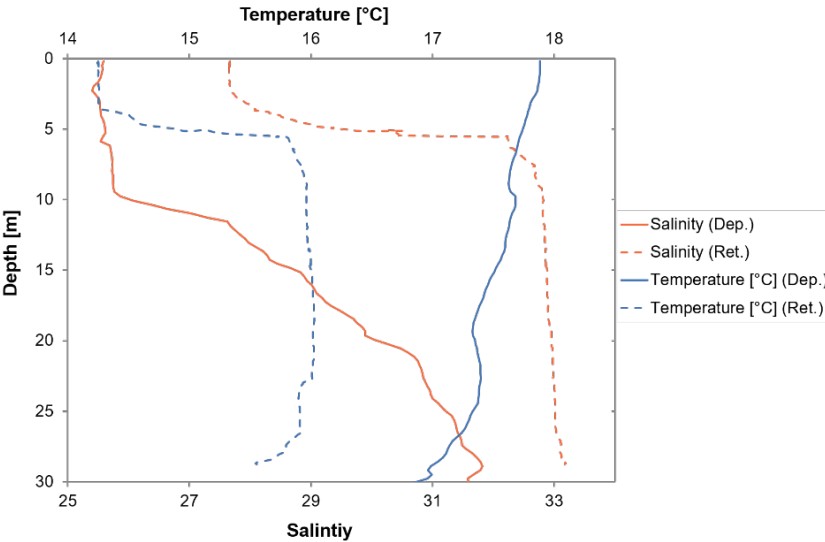


**Figure 5: Salinity and temperature at the time of instrument deployment 28 August 2018 (solid lines) and retrieval 25 September 2018 (dashed lines).**

### 3.1.3 Wake detection rate

The wake detection rate, related to passing distances and vessel force (F), is show in Figure 6. For passages within 3 ship

widths from the instrument (close wake category), the detection rate ranged between 36–100 %, with an average of 57 %. At distances > 3 ship widths, the wake detection was much lower (0–26 %). Due to the low detection rate in the two larger distance categories, only the close wake category will be used in the graphical presentation of how wake depth and longevity

relate to vessel force. Surprisingly, the detection rate of wakes induced by ships passing at distances > 3 ship widths does not seem to be affected by the vessel force, as the percentage of detected wakes is similar for all force bins (Figure 6). Similarly,

the close wake category does not show a clear correlation between vessel force and wake detection rate. However, more passages with large vessel force would be needed to be able to draw any conclusions, since the data is skewed towards lower vessel forces. Nevertheless, the results presented in Figure 6, indicates that passing distance affects the wake detection rate more than the vessel force.

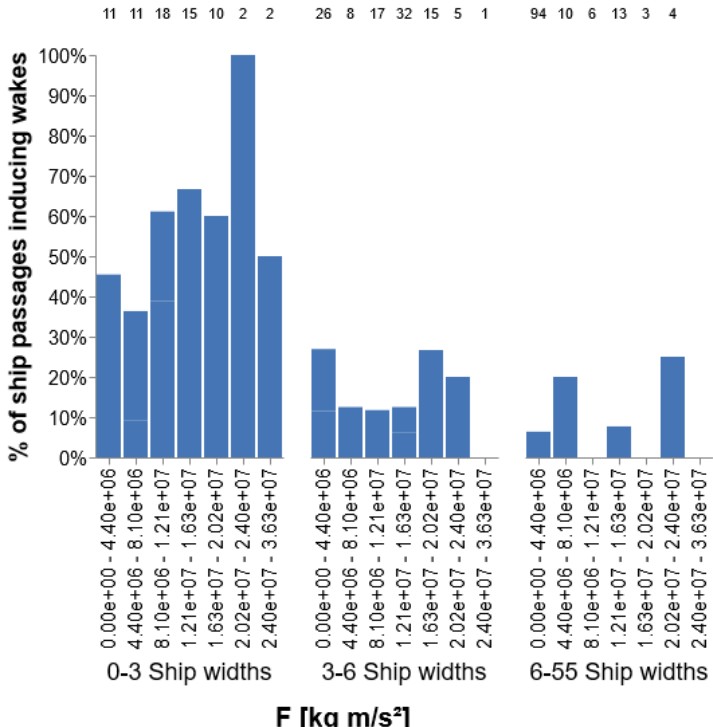


**Figure 6. Wake occurrence for three different categories of passing distances: 0–3, 3–6, and 6–55 ship widths from the instrument. For each distance category, the x-axis shows the force (F) of the passing vessel in Newton. The number above each bar indicate the total number of passages for that category. Note the cut-off in percentage detected wakes at passing distances > 3 ship widths.**

### 3.1.4 Maximum wake depth

The median maximum wake depth for all wakes was 9.5 m (std 4.2 m) for the bubble wake and 11.5 m (std 3.9 m) for the $\varepsilon$ wake (Table 2). The close wake category had larger median values for both the bubble wake and $\varepsilon$ wake, at 11.5 m (std 4.3 m) for the bubble and 13.5 m (std 3.7 m), respectively. These $\varepsilon$ wake depths were not the lower weak rim of the wake, as the threshold values defining the wake region mostly ranged between $10^{-4}$–$10^{-3.5}$ W kg$^{-1}$. These threshold values are really large (e.g. Thorpe (2007)), indicating vigorously turbulent wakes, which probably were homogeneous down to the maximum

depths of the wake region. Previous studies have mainly reported wake depths of 8–12 m (Table 1). In contrast, the results

from this study show that 25 % of the detected bubble wakes were deeper than 12.5 m and 25 % of the $\varepsilon$ wakes were deeper than 14.5 m (Table 2). The deepest detected wakes reached values of 27.5 m for the bubble wakes and 30.5 m for the $\varepsilon$ wake. These values are >10 m deeper than previously reported maximum depths.

In Figure 7, the maximum wake depth is presented for the bubble wake and $\varepsilon$ wake, in relation to vessel force (F). For the bubble wake, the percentage of induced wakes deeper than 12 m increases with increased vessel force, which can be seen in both the close wake category (Fig. 7a) and all single wake passages (Fig. 7b). There is a similar tendency for the $\varepsilon$ wake, although the tendency is more prominent in the figure of all single wake passages (Fig. 7d). However, there was no statistically significant correlation between F and maximum wake depth for either category. The lack of correlation could

partly be explained by the skewed data distribution, as there were few passages with a large F (Figure 6).

**Table 2. Mean, median, maximum value, first quartile (Q25), third quartile (Q75), and standard deviation (std), for wake depth and longevity, for the wake categories; close wakes (0–3 ship widths), single wakes, double wakes, and all wakes in the dataset.**

| Wake category | Bubble wake depth [m] | | | | | | Bubble wake longevity [min] | | | | | |
|---|---|---|---|---|---|---|---|---|---|---|---|---|
| | Mean | Median | Max | Q25 | Q75 | Std | Mean | Median | Max | Q25 | Q75 | Std |
| Close | 11.8 | 11.5 | 27.5 | 9.5 | 13.5 | 4.3 | 00:11:00 | 00:09:59 | 00:28:59 | 00:06:29 | 00:13:15 | 00:06:34 |
| Single | 10.3 | 9.5 | 27.5 | 7.5 | 12.5 | 4.1 | 00:10:14 | 00:08:00 | 00:28:59 | 00:05:29 | 00:13:29 | 00:06:29 |
| Double | 11.2 | 10.5 | 22.5 | 8.5 | 13.5 | 4.4 | 00:12:21 | 00:11:29 | 00:23:29 | 00:07:00 | 00:19:00 | 00:06:23 |
| All | 10.6 | 9.5 | 27.5 | 7.5 | 12.5 | 4.2 | 00:10:50 | 00:08:44 | 00:28:59 | 00:05:53 | 00:15:45 | 00:06:29 |

| | $\varepsilon$ wake depth [m] | | | | | | $\varepsilon$ wake longevity [min] | | | | | |
|---|---|---|---|---|---|---|---|---|---|---|---|---|
| | Mean | Median | Max | Q25 | Q75 | Std | Mean | Median | Max | Q25 | Q75 | Std |
| Close | 13.4 | 13.5 | 30.5 | 11.5 | 14.5 | 3.7 | 00:06:17 | 00:05:59 | 00:13:30 | 00:04:45 | 00:07:44 | 00:02:33 |
| Single | 11.8 | 11.5 | 30.5 | 9.5 | 13.5 | 3.9 | 00:06:22 | 00:05:59 | 00:13:59 | 00:04:59 | 00:07:59 | 00:02:41 |
| Double | 12.9 | 11.5 | 19.5 | 9.5 | 17.0 | 3.8 | 00:09:07 | 00:08:00 | 00:20:00 | 00:06:44 | 00:10:14 | 00:03:53 |
| All | 12.1 | 11.5 | 30.5 | 9.5 | 14.5 | 3.9 | 00:07:08 | 00:06:30 | 00:20:00 | 00:05:00 | 00:08:30 | 00:03:18 |

| | Distance to instrument [m] | | | | | | |
|---|---|---|---|---|---|---|---|
| | Mean | Median | Max | Q25 | Q75 | Std | n |
| Close | 32 | 29 | 82 | 16 | 42 | 21 | 39 |
| Single | 64 | 46 | 184 | 26 | 101 | 51 | 69 |
| Double | 31 | 18 | 120 | 9 | 46 | 32 | 27 |
| All | 55 | 38 | 184 | 16 | 82 | 49 | 96 |

Comparing the median maximum wake depth for the bubble wake and the $\varepsilon$ wake, for the entire dataset, the $\varepsilon$ wake was slightly deeper for all categories (~1 m) (Table 2, Fig. 7). The bubbles in the wake are an indication that surface water has been mixed down at depth and that it has been mixed with the ambient water. The bubbles will remain in the water column, or they can rise or collapse with time, depending on the bubble size. Bubbles with positive buoyancy will have an upward motion counteracting the downward mixing, which could be one explanation to why the bubble wakes are slightly shallower

than the $\varepsilon$ wakes. The dissipation rate of turbulent kinetic energy on the other hand, is a measure of the turbulent motions in the water that mixes the water down. When the turbulence decays, the dissipation also decays and dies out. As the bubbles may remain after the turbulence has died out, it can explain why the bubble wake lasts longer compared to the $\varepsilon$ wake. Another possible explanation to why the $\varepsilon$ wakes are deeper is the calculation method used. The dissipation estimate is influenced by neighbouring cells (Eq. 1) and if there is strong turbulence in one cell and none in the next, the method may

still show some turbulence in the calm cell.

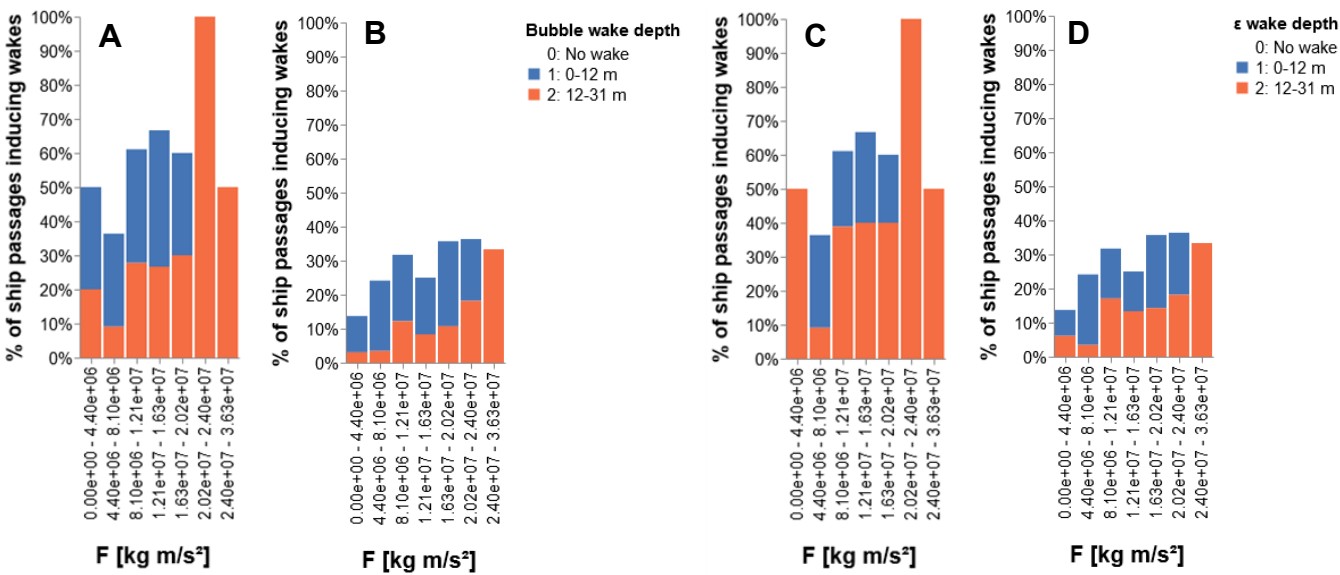

**Figure 7. Maximum wake depth for the bubble wake (a, b) and dissipation rate of turbulent kinetic energy ($\varepsilon$) wake (c, d). (a) and (c) are show the close wakes, induced by ships passing at 0–3 ship widths from the instrument, and (b) and (d) show all wakes induced by single ship passages. The x-axis shows the force (F) of the vessel in Newton. Wake depths within the range presented in 400 previous studies are shown in blue and wakes deeper than previously reported are shown in orange.**

### 3.1.5 Temporal wake longevity

Figure 8 shows the wake temporal longevity related to vessel force, for the same wake categories and parameters as in Figure 7. The median longevity for all wakes was 08:44 min (std 06:29) and 06:30 min (std 03:18) for the bubble and $\varepsilon$ wake respectively) (Table 2). The close wake category had the same longevity for the $\varepsilon$ wake, but the bubble wake was longer at

09:59 min (std 06:34 min). Figure 8 shows no clear correlation between wake longevity and vessel force, for the bubble or $\varepsilon$ wake. Hence, the results from this study, indicate that parameters related to the vessel speed and size do not explain the variation in wake longevity to a very high degree. However, the relatively low number of passages with a large vessel force makes it difficult to draw any definite conclusions without further studies.

In similarity with the maximum wake depth, the double category had a longer duration on average, compared to the single categories, for both the bubble and $\varepsilon$ wakes (Table 2). A majority of the longest wakes (20–30 min) were induced by ships

passing within 3 ship widths of the instrument (Fig. 8). As this indicates that proximity plays a role in the ability to detect the entire temporal longevity of the wake, the close wake category median would be a better estimate of wake longevity, compared to the median longevity calculated from all detected wakes. Compared with previous studies, a detectable signal of the bubble wake from 10 and up to 30 min, is in agreement (Table 1). Furthermore, the timescale of the wake longevity indicates that in highly trafficked areas, where large ships passes every 10–15 min, there is a high potential of a constant influence of ship-induced vertical mixing.

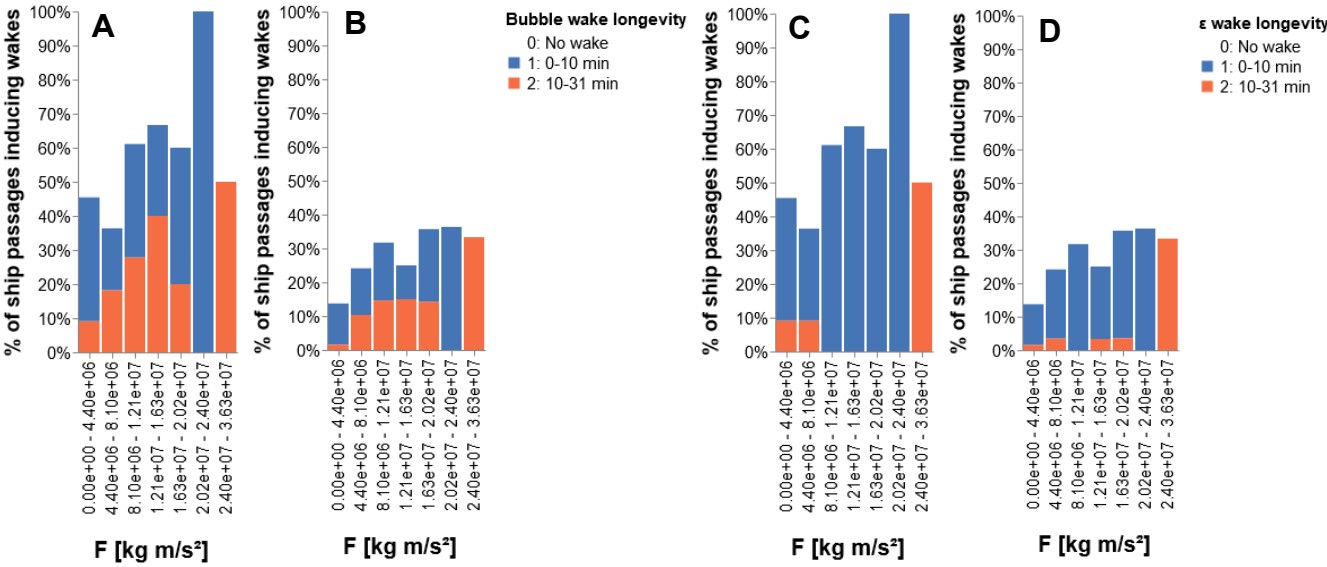

**Figure 8. Wake longevity for the bubble wake (a, b) and dissipation rate of turbulent kinetic energy (ε) wake (c, d). (a) and (c) are show the close wakes, induced by ships passing at 0–3 ship widths from the instrument, and (b) and (d) show all wakes induced by single ship passages. The x-axis shows the force (F) of the vessel in Newton. Wake temporal longevities < 10 min are shown in blue and wake longevities 10–31 min are shown in orange.**

### 3.2 Bornholm satellite image analysis

There was a total of 94 satellite scenes from the period April 2013 to December 2018. Of these scenes, 25 % had a cloud cover of < 23 %, and were analysed for thermal wakes. 48 % of these (n=11) had visible thermal wakes. The monthly distribution of ship passages and occurrence of thermal wakes are shown in Figure 9. As the number of analysed satellite scenes differed between months, the total number of ship passages for each month was divided by the number of analysed scenes. For all months, the majority of the passages did not induce visible thermal wakes. In April-July, there were several induced thermal wakes per scenes (Fig. 9), most of them in May and June. Occasional thermal wakes were found in September and October, but none were found during the winter months (December–February). In the satellite scenes where thermal wakes were visible, and the environmental conditions were right for thermal wakes to be visible, 21 % of the ship passages induced thermal wakes (Table 3). Looking at all the satellite scenes, including those without environmental conditions appropriate for inducing visible thermal wakes, 10 % of the ship passages induced thermal wakes.

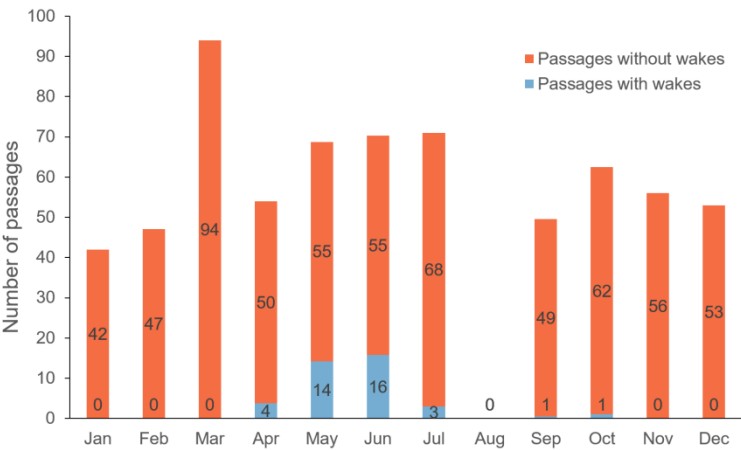


**Figure 9. Seasonal distribution of ship passages for the satellite scenes with < 23 % cloud cover, for the period April 2013 to December 2018. The data labels in the stacked bar indicate the number of passages in each category. As some month has more than one analysed scene, the total number of ship passages for each month was divided by the number of analysed scenes, to get an average number of passages per scene for each month. August had no scenes with < 23 % cloud cover and therefore has no data.**


**Table 3. Number of ship passages in the analysed satellite scenes and the percentage of passages inducing thermal wakes.**

|  | Number of passages | % induced thermal wakes |
|---|---|---|
| Total passages | 1430 | 10% |
| Total passages in scenes with thermal wakes | 684 | 21% |
| Matched thermal wakes | 144 |  |
| Unmatched thermal wakes | 9 |  |

### 3.2.1 Spatial wake longevity

The median length of the matched thermal wakes in the ship lane area was 13.7 km (std 11.8 km), and 25 % were ≥ 20.9 km
(Fig. 10). Assuming that the median speed of the wake-inducing ships in the dataset (13.0 knots) is representative for the ship speed in the area, the calculated temporal wake longevity for the median wake length of 13.7 km was 34 min. The longest thermal wake was 62.5 km, which considering the speed of the wake-inducing ship (20 knots), corresponds to a longevity of 1 h 42 min. In model experiments by Voropayev et al. (2012), the thermal wake signature was still increasing at a distance of 30 ship lengths behind the ship, which would correspond to 6 km for a 200 m long ship. Thus, the thermal wake
length reported in the current study, are up to one order of magnitude larger than previously reported experimental results, indicating an underestimation of thermal wake longevity in previous studies.

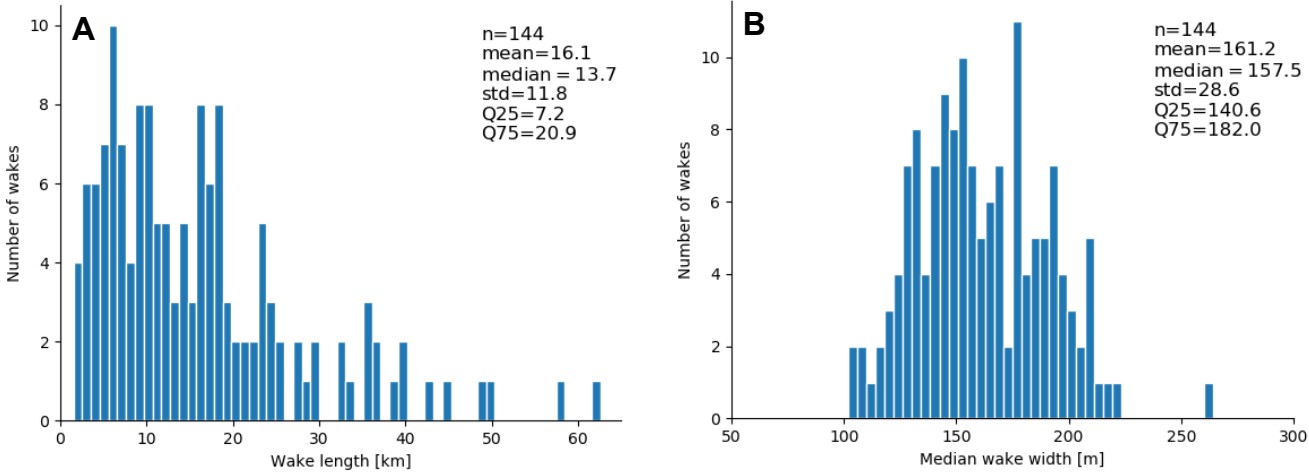

**Figure 10. Wake length (left) and mean width of thermal wakes in the ship lane area, in satellite scenes with visible thermal wakes and < 23 % cloud cover, for the period April 2013 to December 2018 (n=144).**

### 3.2.2 Spatial wake width

The thermal wake width distribution is presented in Figure 10 and Figure 11. The median wake width for the entire dataset was 157.5 m (std 28.6), which is within the range 10–250 m range presented in previous studies (Table 1). The width in this study corresponds to the values presented in Gilman et al. (2011), who also used a ship-based remote sensing approach to estimate width from the visible wake on the sea surface. In contrast, Trevorrow et al. (1994) and Ermakov and Kapustin (2010) reported typical widths of 40–80 m, which is narrower than any widths detected in the current study. However, the last two studies used acoustic measurements of bubbles to estimate the wake width, which could explain the diverging results. The distribution of the median wake width for the different satellite scenes can be seen in Figure 11. Variations in stratification conditions could be one of the explanations to why the thermal wake width varied between scenes. Another reason could be local and regional wind conditions as pointed out in Gilman et al. (2011), or simply the varying temperature gradient between entrained cooler temperatures and warmer temperatures of the upper layer and the resulting exponential adaption process given Newton's law of cooling (Vollmer, 2009; Mallast and Siebert, 2019).

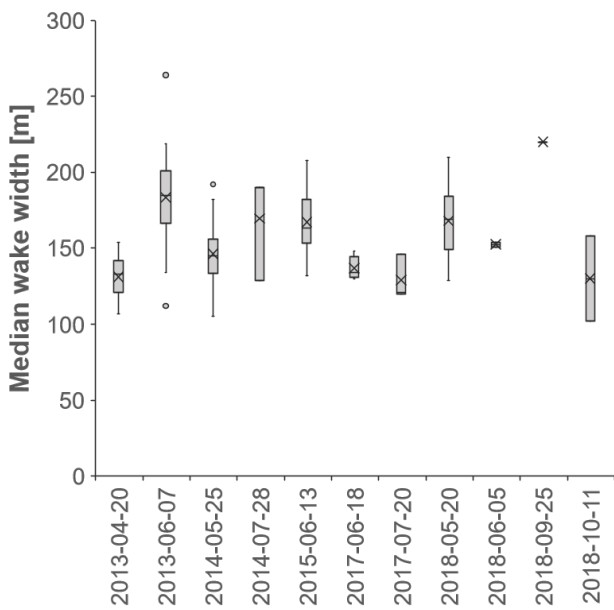

**Figure 11. Median wake width distribution for the thermal wakes in the 11 satellite scenes with visible thermal wakes and < 23 % cloud cover, for the period April 2013 to December 2018. The median values are indicated with an X and outliers with rings (o).**

**3.3 Implications of the spatial and temporal scales of turbulent wakes**

The environmental implications of the spatiotemporal scales of the turbulent wake presented in this study can be illustrated by an example. Using the longevity and width of the "median" turbulent wake, it is possible to estimate the area of the ship lane being affected by the turbulent wake at any given time. The traffic separated ship lane in the sound north of Bornholm is intensely trafficked, with 50,000 ship passages every year (HELCOM, 2010). A typical example of the number of ships

present in the area at any given time can be seen in Figure 12, which shows the ships with AIS transmitters present in the sound at the time of the satellite scene from 2014-05-25 10:01. The ship lane area and the traffic separation zone are indicated, with each ship lane being 5 km wide and approximately 30 km long, which correspond to an area of 150 km$^2$. During the time of the satellite passage, there were four ships present in both the south-east ship lane in the traffic separated part in the Bornholm sound, and in the north-west ship lane. The median thermal wake length (13.7 km) and width (157.5 m)

(Fig. 9) gives an average thermal wake area of 2.16 km$^2$. Consider a scenario where all wakes are uniformly distributed without overlap. With no overlapping wakes, four median ship wakes would cover an area of 8.6 km$^2$. With a ship lane area of 150 km$^2$, the area covered by thermal wakes would correspond to 5.8 % of the ship lane. Considering the frequent ship traffic in the Bornholm sound and the satellite observations in this study, the presence of eight ships in each separation zone at the same time should occur frequently. In this case the covered area would be 17.3 km$^2$, which corresponds to 11.5 %. In

Figure 12, some of the wake tracks are overlapping, thus it does not fully correspond to the scenario of uniformly distributed wakes. However, the figure still gives a conceptual visualisation of how large the part of the ship lane area is that can be influenced by thermal wakes.

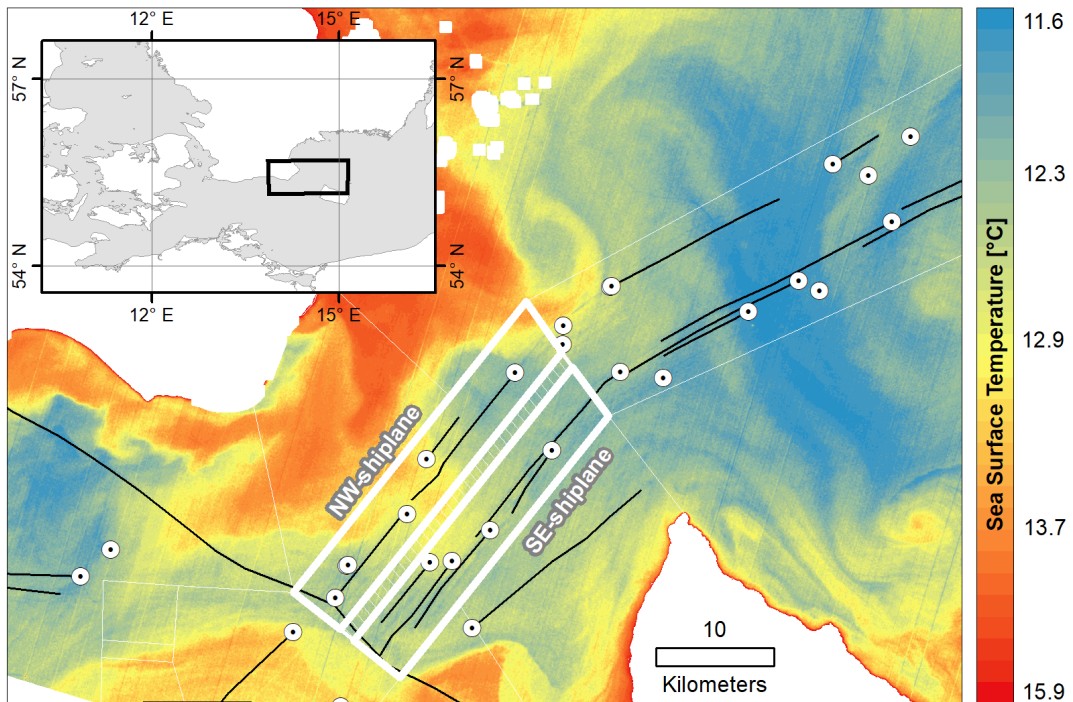

**Figure 12. Ships and visible thermal wakes in the Bornholm sound from the analysed satellite scene from 2014-05-25 10:01. White circles with black dots indicate ships with an AIS transmitter present in the area at the time of the satellite passage, and the black lines are the digitalised visible thermal wakes from the satellite scene. White lines indicate the ship lane area, the bold lines mark the North-West and South-East ship lanes, and the dashed area is the traffic separation zone. Landsat-8 image courtesy of the U.S. Geological Survey.**

In addition to an estimate of the area affected by the turbulent wake, it is also possible to consider the frequency at which the water mass in a certain point would be influenced by a turbulent wake. An average of 50,000 ship passages in the Bornholm sound, corresponds to 25,000 passages in each direction, which divided over a year would correspond to approximately one passage every 21 min (~ 3 per hour). Consider a scenario, where instead of a uniform distribution of ships in the entire ship lane, all ships travel along the exact same path. The calculated median temporal thermal wake longevity for the satellite data was 34:00 min. As the thermal wake longevity is longer than the average time between ship passages, the assumption that all ships travel the exact same route would mean that the water mass along the travelled route would be under constant influence of a ship-induced thermal wake. Now consider the same scenario, but using the median temporal longevity for all the ADCP wake measurements from the close wake passages, 09:59 min for the bubble wake and 05:59 min for the $\varepsilon$ wake (Table 2). As the bubbles in the turbulent wake are visible for 09:59 min, the assumption that there is a ship passage every 21 min means that there is 11min between each ship passages when there are no bubbles. If using the median temporal longevity for the $\varepsilon$ wake instead, the time would be 15 min. Hence, the bubble wake would influence the water mass in a certain point every 11 min and the $\varepsilon$ wake every 15 min.

The difference in temporal longevity, between the ADCP measurements and satellite observations, can partly be explained by the fact that the two methods measures different aspects of the turbulent wake. The ADCP measurements show the very turbulent core of the wake. The dissipation rate of turbulent kinetic energy ($\varepsilon$) gives an estimate of the intensity of the mixing, and both the $\varepsilon$ and bubble wake gives an estimate of the spatial scales of the turbulent wake. The satellite observations, on the other hand, show the thermal signal of the water mass that has been produced by the turbulent mixing. The mixed water from the turbulent wake will remain even after the turbulence has died away, and is a measure of water that has been influenced by mixing. Hence, both methods can be used to estimate the spatial and temporal scales of ship-induced mixing, but the ADCP measurements give an estimate of the turbulent wake, and the satellite analysis shows the scales of the water influenced by the turbulent wake.

The above calculated area coverage of thermal wakes, and the frequency at which the water mass in a certain point would be influenced by ship-induced mixing, represents two extremes. The first scenario assumes a uniform distribution of all ship wakes, and the second scenario assumes that all ships travel along the same route. However, in reality some of the wake regions would be overlapping, and most ships would travel similar, but slightly different routes in the ship lane. Nevertheless, based on the results presented in this study, areas like the Bornholm ship lane in the Baltic Sea could be considered under a near constant influence from ship-induced turbulent mixing. Even if the water column regains its stratification quite quickly, the mixing of the wake water with the surrounding water would take much longer. In a natural marine system, the water column is often stratified due to surface heating and/or freshwater influence. The wake turbulence interacts with this stratification by mixing the water and entraining deeper waters into the wake. The stratification may, in turn, reduce the vertical extent of the wake relative to what it would have been in a homogeneous water column (e.g. Voropayev et al. (2012)). During periods of seasonal stratification, nutrients in the surface layer are depleted, and the supply of nutrients from below is limited due to damping of the vertical mixing by the stratification (Reissmann et al., 2009; Snoeijs-Leijonmalm and Andrén, 2017). In coastal regions, nutrients can be brought up to the upper mixed layer by coastal upwelling, but in the open ocean, the nutrient supply is dependent on vertical mixing (Reissmann et al., 2009). If the vertical mixing is intense and deep enough, it will bring up nutrient rich water from below the stratification to the upper surface layer, which can increase primary production and sustain algal blooms. In ocean systems unaffected by human activities, vertical mixing in the surface layer is induced by wind, and the depth of the mixing depends on the wind strength and duration, as well as the input of buoyancy from heating and fresh water (Thorpe, 2007). In temperate oceans like the Baltic Sea, the seasonal stratification occurs during the summer season, which is also the period with the least wind (Reissmann et al., 2009). Thus, in unaffected seasonally stratified waters, there is little vertical mixing during the summer months. However, in areas with intense ship traffic there is a frequent input of ship-induced vertical mixing. In the Baltic Sea, at any given moment, there are circa 2000 moving vessels (HELCOM, 2010). A scoping calculation based on the average main engine power and velocity per ship type presented in Jalkanen et al. (2014), and the distance travelled by each ship type from

Hassellöv et al. (2019), will give a yearly input of turbulent kinetic energy from ship wakes of 3.9 GW. Based on the total surface area of the Baltic Sea (including Kattegat and Skagerrak) and using the conservative assumption that the ships are running at 50 % Maximum Continuous Rating (MCR) (Buhaug et al., 2009; Smith et al., 2015), the average energy input from turbulent ship wakes would be 0.0044 W m$^{-2}$. This ship-induced turbulent kinetic energy will mostly dissipate, but a certain fraction will be used to mix the water column in case of stratified water. This is to be compared with the dissipation rate of turbulent kinetic energy caused by wind and wave generated turbulence. Below the direct wave breaking layer, about one wave height thick (e.g. Sutherland and Melville (2015)), the dissipation rate of turbulent kinetic energy follows the "law of the wall" (Thorpe, 2007). There, the integrated dissipation rate of turbulent kinetic energy between the depths $z_1$ and $z_2$ can be written as Eq. (4):

$$\rho_0 \frac{u_*^3}{\kappa} ln \frac{z_2}{z_1} \qquad\qquad (4)$$

where $\rho_0$ is the water density, $\kappa$ is the von Kármán constant ($\cong 0.4$), and $u_*$ is the friction velocity. The friction velocity can be estimated from the wind velocity at 10 m height ($U_{10}$) as Eq. (5):

$$u_* = \sqrt{\frac{\rho_a}{\rho_0}} C_D U_{10} \qquad\qquad (5)$$

where $\rho_a$ is the air density, and $C_D$ is a drag coefficient. An estimate of the integrated wind generated dissipation rate at Gotska Sandön in the Baltic Sea between 1 and 20 m depth gives 0.002 W m$^{-2}$ in summer time and 0.007 W m$^{-2}$ in wintertime, based on wind observations and using the parameterization of Smith (1988) for the drag coefficient. The dissipation rate of turbulent kinetic energy caused by vessels is therefore one order of magnitude larger than that caused by winds during summer at the depths where the turbulence may cause mixing of the seasonal thermocline. That is when averaged over the whole basin. The local impact in shipping lanes and behind individual ships is much larger. In the Baltic Sea, the seasonal thermal stratification is located at 10–20 m depth (Stigebrandt, 2001; Leppäranta and Myrberg, 2009), and in many of the areas where the major ship lanes are situated, the median water depth is between 20–50 m (Jakobsson et al., 2019). Consequently, during summer stratification, ship-induced turbulent mixing has a large potential to alter gas exchange and nutrient availability on a local/regional scale, which should be considered when evaluating environmental impact from shipping.

The results presented in this study, also have implications for monitoring and data collection in areas with ship traffic. An especially interesting example are the so called FerryBox systems, which are placed on ships and conduct continuous measurements of parameters such as $O_2$ concentration, salinity, temperature, and sometimes also $pCO_2$, Chlorophyll a, and pigments (Petersen, 2014). There are currently seven passenger ferries equipped with FerryBox systems in the Baltic Sea, which are traveling along the major shipping lanes all or part of the journey (https://www.ferrybox.com/routes_data/routes/baltic_sea/index.php.en). The intake of water is from an inlet in the ship hull (Petersen, 2014), which would correspond to somewhere between 2–10 m depth. Considering the wake longevity of the

thermal and turbulent wakes presented in this study, there is a high likelihood that a ship traveling in a major ship lane, could be in the wake of another ship. In that case, the water being measured by the FerryBox is the water of the turbulent wake, and thus not representative for the conditions outside the ship lane. The validations made for FerryBox measurements are being made using the same water source as the FerryBox (Karlson et al., 2016), which would still be part of the ship lane area, and not the unaffected waters outside the ship lane. As the measured temperature differences between inside and outside the thermal wakes, was up to 1°C in some of the scenes (see Fig. 4 for example), and as the bubbly wake affects gas exchange and saturation, it is important to know if the measurements are affected by ship-induced turbulence. Hence, the effect of ship-induced vertical mixing should be considered when using data collected from FerryBox systems.

Among the ADCP measurements, there were a few wakes which reach depths of >18 m (Table 2). The deepest wake in this dataset was induced by a cargo ship with a beam of 25 m, length of 229 m, and draught of 7 m. It passed the instrument at a distance of 34 m and a speed of 19 knots. The cargo ship had a Gross Tonnage similar to the average of container and Ro-Ro cargo ships in the Baltic Sea (HELCOM, 2018), indicating that ship-induced mixing to depths of 30 m could be a common, but undetected occurrence. The hypothesis that vertical mixing of this depth could be more frequent than expected from previous studies (Table 2) is supported by the fact that similarly sized ships passing at the same distance as cargo ship inducing the deepest wake, also induce mixing to depths greater than 15 m. The lack of previous reports of vertical mixing of this magnitude can partly be explained by the fact that no previous study has targeted this specific research question. Moreover, measurements made using similar methods, but for other purposes, are seldom conducted in ship lanes and particularly not from below. On the other hand, the difference in wake depth for ships of similar size and passing distance could also be due to differences in stratification, as a strong stratification can dampen the vertical development of the wake (Kato and Phillips, 1969). During the ADCP measurement campaign, water column stratification was measured at deployment and retrieval of the instrument (Fig. 4). Three hours before the instrument retrieval, a cargo ship passed at a distance of 21 m and induced a bubble wake of 13.5 m depth and a $\varepsilon$ wake 17.5 m depth. At the point of retrieval, the CTD measurement showed a strong thermal stratification at 5 m depth. At the time of ship passage, intense vertical mixing induced the wake down to 17.5 m depth, and the likelihood that the thermal stratification at 5 m depth was present during the longevity of the wake, is small. This means that the stratification was influenced by the wake and that waters above and below the thermocline were mixed with each other. Three hours later the water had re-stratified and the mixed water has spread out laterally. However, during the longevity of the wake, the stratification was most likely strongly affected, and mixing between water masses would occur. The effect of the mixing, in terms of changes of the physical and chemical characteristics of the water mass, is irreversible. However, the effect of the mixing event that remain after re-stratification (i.e. changes in chemical composition, gas exchange, temperature), are not possible to observe with the methods used in this study. Still, ship-induced turbulence affects the local and regional stratification, even though the contribution from each single ship is difficult to observe with an ADCP after the water has re-stratified. Nevertheless, further studies are needed to determine the impact of stratification on the vertical development of the turbulent wake, and how it varies with the ship's

draught and speed. Thus, the results from this study shows that vertical mixing to depths 30 m occurs, and possibly at a high frequency. However, as the current knowledge about the wake distribution is poor (especially on a vertical scale), and further studies are needed to determine when, and at which frequency, vertical mixing reaching this depth occurs.

**3.4 Limitations and Future outlook**

The measurements in this study indicated resuspension and turbulence at the sea floor at 30 m depths, induced by the Kelvin wake from passing ships. In shallow water regimes the waves of the Kelvin wake give rise to increased current speeds at the sea floor, which can lead to resuspension (Soomere and Kask, 2003; Soomere, 2007). These effects were seen at quite large distances from the passing point, which is expected as the Kelvin wake travels along the sea surface and have a larger spatial

extent compared to the turbulent wake. The effect and waves of the Kelvin wake are comparable to swells or wind waves, but their temporal extent is much shorter and the wave characteristics (wave period, significant wave hight) can be different from the natural wave regime. In the Baltic Sea, Danielsson et al. (2007) have shown that wind waves are important for sediment resuspension at more than 40 m depth, which is comparable to the results in this study. However, the effect of Kelvin wakes was outside the scope of the current study but has been investigated by (Soomere and Kask, 2003; Soomere,

2007); Soomere et al. (2009). Nevertheless, the observations indicate the importance of including the effect of the Kelvin wake where shallow water regimes apply, when estimating the environmental impact on the marine environment in intensely trafficked ship lanes.

The lack of detectable thermal wakes in the satellite dataset during the winter months was expected. A thermal stratification

is needed to entrain cooler water from below, induced by the turbulent wake and cause a surface temperature gradient. The Bornholm region usually has a no thermal stratification during winter (Reissmann et al., 2009; van der Lee and Umlauf, 2011). Therefore, the method of estimating the spatiotemporal scales of the turbulent wake using satellite SST measurements is limited to seasons and regions where strong thermal stratifications occur. Moreover, the low percentage of available satellite scenes with little enough cloud cover makes alternative remote sensing techniques, such as drones, a possible

alternative. Drones could also be used for a longer time period in the same area and in combination with under water measurements.

When comparing the observations from the satellite data and the ADCP measurements, remember that they were obtained in different ocean basins and during different stratification conditions. As stated in the methods section, the sites were chosen

based on the needs for the two types of measurements. The frequent ship traffic and a separation zone in the Bornholm ship lane made it suitable for detecting the longevity and occurrence of thermal wakes. However, the intense maritime activity in the area and the larger depth, made it both riskier and logistically difficult to place ADCP instruments in the Bornholm ship lane. The authors have previously lost two instruments in the Bornholm area, due to maritime activity, which is why it was considered unsuitable for a longer measurement period. Instead, the Gothenburg area was chosen, as it has a varied and

intense ship traffic, with several ferries and cargo vessels on route, ensuring daily passages. It also gave the possibility to access detailed draught information from the ships from the Port of Gothenburg, for the ships visiting the port. The Gothenburg site was not considered suitable for the satellite study, due to the lower amount of ship passages per day and cloudier weather conditions. These circumstances resulted in the decision to use different locations for the two studies. Nevertheless, satellite images were retrieved for the Gothenburg site for the *in situ* measurement period, but they were too

cloudy to be usable for any analysis. For future studies focused on characterising the development of the turbulent wake, the ideal would be to make remote sensing and ADCP measurements simultaneously at the same site. However, it would probably be more suitable to use drones instead of satellite images, as a drone is more flexible and makes it possible to operate during cloudy conditions, to capture the development in time, and to use both static and dynamic approaches when documenting the wake.


In addition to the difference in geographical area, the satellite observations show a snapshot of the ocean surface, whereas the ADCP instrument does not measure the top 4 m of the water column. Hence, the two methods never capture the same part of the wake, which could lead to different results using the two methods. Moreover, the satellite observations show the effect of mixing, while ADCP observations show the actual turbulence that causes the mixing. After the mixing has

occurred, the mixed water may move outwards – a movement not causing enough turbulence to be seen by the ADCP. This could be one explanation to why the thermal wake longevity is longer, compared to the ADCP wake longevity. Furthermore, the thermal wake being a proxy for the effect of the mixing and not the turbulence itself, is also one of the reasons to why the ADCP results have not been used to estimate a thermal wake depth. However, the large variation in vertical and horizontal distribution of the turbulent wakes observed during the wake analysis (inferred by comparing the signal between the slanted

ADCP beams), strongly indicate that the vertical cross section across the width of the turbulent wake is non-uniform and varying. Based on these observations, the vertical cross section of the thermal wake is most likely also non-uniform and will differ in dept along the cross section. However, there is a need for further studies to clarify how the ship design, speed, and propeller (number and rotational direction), interact with water column stratification and currents, in forming the "shape" of the turbulent wake.


The satellite analysis showed a median wake width of 157.5 m (Fig. 10), from which it would be expected to frequently detect wakes from ships passing up to 75 m from the instrument. The ADCP results indicate a similar range for frequent detection, namely 50 m. When discussing the detection range of the ADCP instrument, the influence of currents and wind should be considered. In this study, the water speed and waves were measured with the ADCP, and the wind effect on

currents and waves were considered captured by those measurements. As a current can move the wake towards or away from the instrument, the current speed and direction must be taken into consideration when estimating at what distance from the ship a wake is likely to be detected. Trevorrow et al (1994) conducted measurements within 2–5 m of the turbulent wake and reported difficulties in catching the bubble signal from the wake using vertical sonars, as the wake often drifted out of the

sonar range before it had completely dissipated. In this study, a majority of the passages (50–60 %) occurred when there was a weak or no current at the position of the ADCP instrument (data not shown). Moreover, a current speed towards the instrument did not increase the likelihood of detecting the wake, especially not when ships passed further away from the instrument (data not shown).

In addition to the currents, the width and size of the ship should also be taken into consideration when discussing detection related to the passage distance from the instrument. The distance between the ADCP instrument and ship, is calculated from the position of the AIS transmitter. As the transmitter is often located at the middle of the ship and, a wide ship might be passing right over the instrument even though the AIS stamp indicates that the ship is 25 m away. Thus, larger ships are possibly closer to the instrument than what is registered by the AIS, which could potentially influence the wake detection. To adjust for this bias, the graphical presentation of the data has the distance to the instrument has presented in ship widths instead of meters. A large majority of the ships inducing wakes in the ADCP measurements were 20 m or wider, and the wider ships were overrepresented among the passages inducing wakes, comparing the wake width for the entire dataset. Moreover, the smallest ships (width < 10 m) rarely induced wakes, and then only when passing within 75 m of the ADCP. A similar pattern can also be seen when looking at the length of the ships inducing the wakes.

In the current study, the water column stratification was only measured at deployment and retrieval of the instrument, hence the importance of stratification could not be included in the analysis of this study. However, the presence and strength of the stratification will influence how much turbulence that is required to mix water and substances across the thermocline (e.g. Kato and Phillips (1969)). In a stratified fluid, vertical mixing removes energy from the turbulence, reducing the vertical extent of the wake development. Stratification will also cause mixed fluid to spread out laterally, which causes an adjustment of the wake stratification to the surrounding stratification, resulting in a widening of the wake as well as an additional limitation of the vertical extent (Voropayev et al., 2012). As the aim of the current study was to present an order of magnitude estimation of the spatial and temporal scales of the turbulent wake, the lack of stratification measurements does not present at problem within the current scope. However, for future studies with the aim of characterising the development of the turbulent wake and quantifying the ship-induced vertical mixing, stratification measurements will be necessary in order to understand the interaction between the stratification and the turbulent wake. Moreover, as the stratification must be expected to be an important factor for wake depth, it could be one explanation for the absence of statistically significant correlations between wake depth and vessel force.

In order to determine when vertical mixing reaching depths of 30 m occurs, and how common it is, future studies need to simultaneously measure the wake in more than one point, in order to get the cross section of the wake. One way of achieving this would be to conduct measurements with several ADCPs placed on a row perpendicular to the ship lane. This would give a cross-section of the wake, which could be used to describe both the width and depth of the turbulent wake. As the

measurements in this study were made using one instrument, only the depth of the wake could be measured, and only at one point in the wake cross-section. Moreover, a line of instruments would also be able to capture a drifting wake and thus better estimate the true longevity. One of the limitations of the longevity estimation in this study, is that currents could potentially shift the wake away from the instrument. Using multiple instruments would increase the chance of capturing the entire wake development, as it would cover a larger area, thus increasing the reliability of the longevity estimation. As the results from this study indicate that proximity is of importance for detecting the turbulent wakes using ADCP measurements, multiple instruments would increase the area where ships can pass close to the instrument. In addition, if the maximum depth of the wake is located only in a certain region of the turbulent wake, the likelihood of measuring that part of the wake is small when only one instrument is used. This spatial limitation of the current study makes it difficult to determine if the small number of detected deep wakes was because of low occurrence, or because using only one instrument made it difficult to successfully measure the deepest part of the wake. Thus, multiple instruments would increase the ability to identify when and where the very deep mixing occurs and shed further light upon how frequently deep mixing is induced. It would also be beneficial to conduct concurrent measurements using ADCPs and remote sensing. In the current study, the satellite analysis and ADCP measurements have been conducted at different locations and time periods, but concurrent measurements would give a more complete picture of both the large horizontal temporal and spatial scales, as well as the vertical scales.

## 4 Conclusions

Based on a large sample of *in situ* measurements, the median spatiotemporal extent of turbulent ship wakes have been estimated to a depth of 13.5 m and longevity of 09:59 min, based on ADCP measurements. Thermal wake width and longevity have been estimated to a median of 157.5 m and 13.7 km respectively, based on SST satellite image analysis. The results show frequent detection of turbulent wakes deeper than 12 m, which is deeper than previously reported. Moreover, in areas with intense ship traffic, the presented temporal and spatial extent of the turbulent wakes are of a scale relevant to consider when assessing environmental impact from shipping.

## 5 Data Availability

Acoustic measurement data available upon request for non-commercial purposes. AIS data available through HELCOM according to their data policy. Satellite images freely available at https://s3-us-west-2.amazonaws.com.

## 6 Author contribution

I-M. Hassellöv, A. T. Nylund, L. Arneborg and A. Tengberg conceptualised and conducted the *in situ* field measurements and consecutive analysis and visualisation. A. T. Nylund developed the code used in the analysis, with contribution from L.

Arneborg. U. Mallast conducted the data curation and formal analysis of the satellite images, with contribution from A. T. Nylund. The manuscript was prepared by A. T. Nylund with contributions from all co-authors.

## 7 Competing interests

The authors declare that they have no conflict of interest.

## 8 Acknowledgements

Acknowledgment of funding for the OCEANSensor project by the Research Council of Norway (project number 284628) and co-funding by the European Union 2020 Research and Innovation Program, as part of the MarTERA Program. Acknowledgement to the Swedish Institute for the Marine Environment (SIME), for supplying the AIS dataset. This work has been partially supported by MarineTraffic, by the use of their database of vessel information.

## 9 Abbreviations

ADCP - Acoustic Doppler Current Profiler

AIS - Automatic Information System

ANOVA - Analysis of Variance

EOF - Empirical Orthogonal Function

HELCOM - Baltic Marine Environment Protection Commission

IMO - International Maritime Organization

MCR - Maximum Continuous Rating

MNDWI - modified normalized difference water index

NDRC - National Defense Research Committee

SIME - Swedish Institute for the Marine Environment

SMHI - Swedish Meteorological and Hydrological Institute

TIRS - Thermal Infrared Sensor

TOA - top-of-the-atmosphere

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
