# Peer review of "In situ* observations of turbulent ship wakes and their spatiotemporal extent"

_Ocean Science, 2020_

## Referee Comment (RC1) · Anonymous Referee #1 · 19 Aug 2020

General comments. The paper is mostly descriptive and practically no physical analysis of the observations is performed. It looks mostly like a report on some routine observations, like "a pile of data", and only methodological aspects of the work are discribed, although not clearly enough (see, some specific comments below). One cannot find in the text any new physical effects. The paper in its present form does not look interesting and informative from a scientific point of view. The aim/motivation of the paper is not clear.

Specific comments. 1. An error in formula (1) 2. A scheme of the ADCP deployment and recording of ship wakes has to be presented to understand how the ship wakes

are recorded by the ADCP. For instance, the bubble wake manifestations similar to one in Fig.2 appear when the ADCP is towed across the ship wake, or if the wake is moving in the cross wake direction due to currents passing by the ADCP beams. How thus the record of a ship wake in Fig.2 could be obtained for a stationary looking upward ADCP? Was that due to a current moving a wake through a zone illuminated by ADCP? 3. Why the wakes appeared in the ADCP records for ships passed by at some distances from the ADCP? Because of the wake turbulent diffusion? If so, why not to analyze, e.g. the characteristic times of the turbulent diffusion, the diffusion spatial/temporal decay, etc.? 4. Line 301. I cannot understand how this can happen : " . . . . . .when two ships passed the instrument at the same time" 5. Categorization of the ships in the context of their turbulent wakes does not look physically justified. More reasonable would be to relate the wakes to the ship weight, draught, speed, possibly to the size/number of propellers. 6. line 331 "As the fraction of detected induced wakes at similar distances differ between ship types, it is an indication that the ship type impacts the characteristic of the turbulent wake" . I disagree with the statement and I think that the difference is determined mostly by the ship weigh and ship speed. 7. The paper is full of obvious, trivial statements, e.g. "in general the deepest wakes were caused by ships passing closer to the instrument, whereas ships passing at larger distances from the instrument (100–199 m) mainly caused shallower wakes . . ." (lines 369-370) "the maximum dissipation rates . . . in the core of the wake . . ..are . . ...much larger than what is usually observed in the core of, or below, the surface mixed layer" (lines 403-405), etc. etc.

---

## Referee Comment (RC2) · Anonymous Referee #2 · 3 Sep 2020

Overview:

The main finding of this study is that turbulent ship wakes can reach deeper than the previously observed values, however, no physical explanation or discussion is provided. It is expected that ship specifications and speeds determine ship wake depths, so the authors should be able to discuss further based on the available ship information.

It is not clear why the authors chose two separate locations for the in-situ and remote sensing study. Is it possible to find satellite imagery for the in-situ measurement period?

Also, vertical profiles should have been measured more frequently to see the effect of

wakes on stratification and mixing.

Specific comments:

Line 191: Capitalize Python

Line 224: Why 15%? Please justify.

Line 355-359: Why do the bubble wakes look different from turbulence? Please discuss further.

Line 550-552: This seems to be a negative result: the stratification was not affected by the wake. Remove this part. I suggest that the authors measure more vertical profiles in the study area and/or provide literature for more data to characterize general and unusual environmental conditions. 4 casts x 2 days are not enough.

Line 574-581: As mentioned above, please try to find satellite imagery that covers the in-situ measurement area.

Line 587: Note that winds are also important.

Line 618: What parameters?

---

## Author Comment (AC1) · 16 Oct 2020

The comment was uploaded in the form of a supplement:
https://os.copernicus.org/preprints/os-2020-59/os-2020-59-AC1-supplement.pdf

---

## Author Comment (AC2) · 16 Oct 2020

**Answers to RC 2**

***Overview:*** *The main finding of this study is that turbulent ship wakes can reach deeper than the previously observed values, however, no physical explanation or discussion is provided. It is expected that ship specifications and speeds determine ship wake depths, so the authors should be able to discuss further based on the available ship information.*

We acknowledge the comment to relate the wake depth and longevity to another parameter than ship type, and therefore suggest a revision of the result section. It is beyond the scope of this paper to investigate the dependence between various non-dimensional parameters, which would be the most physically justified thing to do. However, we expect that the wake size to a large degree depends on the force or power put into the water by the propeller. We do not have data on these parameters, but we do expect that both the force and the power depend on the dimensions and speed of the vessel though water. The drag force on the ship is one of the possible resistances the ship is exposed to.

We therefore propose exchanging the current figures 5 and 6, to Figure 1 and Figure 3 and/or Figure 2 and Figure 4 below. The new figures show the wake depth and longevity in relation to force (F), calculated as $\rho * ship\ width * ship\ draught * ship\ speed^2$ [kg m s$^{-2}$], with seawater density ($\rho$) equal to 1025 kg m$^{-3}$. This parameter is proportional to ship drag and will relate the wake depth and longevity to vessel size and speed, which we agree are parameters expected to have an impact on the formation of the turbulent wake.

[Figure]

*Figure 1. Maximum wake depth for the bubble wake (a) and dissipation rate of turbulent kinetic energy (ε) wake (b), for the wakes induced by ships passing at 0-3 ship widths from the instrument. The x-axis shows the force (F) of the vessel in Newton, calculated as ρ\*ship width\*ship draught\*ship speed$^2$. Wake depths within the range presented in previous studies are shown in blue and wakes deeper than previously reported are shown in orange.*

Figure 1 and Figure 2 shows the maximum wake depth for the bubble wake and dissipation rate of turbulent kinetic energy (ε) wake. The difference between the figures is Figure 1 only shows the ships passing within 0-3 ship widths from the instrument (roughly corresponding to 75 m), whereas Figure 2 shows all the passages in the dataset. As a majority of the induced wakes are from ships passing within 0-3 ships widths from the instrument, we propose to limit the graphical presentation of the dataset to this part of the dataset. We can see a clear cut-off in the percentage of detected

wakes at 3 ship widths. Hence, as we are currently not able to correct for the uncertainties introduced by the distance factor (see discussion in manuscript for further details), we argue that the closer passages give a better representation of the actual temporal and spatial scales of the turbulent wake, than including the entire dataset. Nevertheless, we propose presenting statistics for both the entire dataset and the passages within 3 ship widths of the instrument.

[Figure]

*Figure 2. Maximum wake depth for the bubble wake (a) and dissipation rate of turbulent kinetic energy (ε) wake (b), for all single passages. The x-axis shows the force (F) of the vessel in Newton, calculated as seawater density\*ship width\*ship draught\*ship speed[2]. Wake depths within the range presented in previous studies are shown in blue and wakes deeper than previously reported are shown in orange.*

[Figure]

*Figure 3. Wake longevity in minutes for the bubble wake (a) and dissipation rate of turbulent kinetic energy (ε) wake (b), for the wakes induced by ships passing at 0-3 ship widths from the instrument. The x-axis shows the force (F) of the vessel in Newton, calculated as seawater density\*ship width\*ship draught\*ship speed[2]. Wake temporal longevities < 10 min are shown in blue and wake longevities 10–30 min are shown in orange.*

Similarly, Figure 3 and Figure 4 and shows wake longevity for the bubble wake and ε wake, for the ships passing within 0-3 ship widths from the instrument and all single passages in the dataset, respectively. We suggest the same presentation and statistics as for the maximum wake depth

parameter. For both wake depth and longevity, we suggest including the figure with the closest passages in the manuscript, and the figure for the entire dataset to be added to a supplementary info. If requested, the supplementary info can also include a figure showing the wake detection for different distance categories, to illustrate the wake detection cut-off at 3 ship widths (Figure 5).

[Figure]

*Figure 4. Wake longevity in minutes for the bubble wake (a) and dissipation rate of turbulent kinetic energy (ε) wake (b), for all single passages. The x-axis shows the force (F) of the vessel in Newton, calculated as seawater density\*ship width\*ship draught\*ship speed[2]. Wake temporal longevities < 10 min are shown in blue and wake longevities 10–30 min are shown in orange.*

[Figure]

*Figure 5. Maximum wake depth for the dissipation rate of turbulent kinetic energy (ε) wake. The data is presented for three different categories of passing distances: 0-3, 3-6, and 6-55 ship widths from the instrument. For each distance category, the x-axis shows the force (F) of the vessel in Newton, calculated as ρ\*ship width\*ship draught\*ship speed[2]. Wake depths within the range presented in previous studies are shown in blue and wakes deeper than previously reported are shown in orange. Not the clear cut-off in detected wakes at passing distances > 3 ship widths.*

In addition to the change in figures, we also propose a change to table 2 and 3 in the manuscript. We will remove the ship type category statistics and instead include statistics for the close passage category (0-3 ship widths) (Table 1). As mentioned above, the double wakes will not be included in the figures, as we cannot determine which vessel that induced the wake, and thus lack the necessary

vessel information to do the calculations. However, we argue that it is relevant to include these wakes in the statistical analysis, as the double category constitutes 28 % of the detected wakes. The aim of the paper is to describe the temporal and spatial scales of the turbulent wake, and the double passages are one type of wakes that frequently occur in the dataset. The inability to include them in the figure is not related to any uncertainty of the wake measurement and should therefore be included in the overall analysis.

*Table 1. Mean, median, first quartile (Q25), third quartile (Q75), standard deviation (std), minimum value, and maximum value for wake depth and longevity for the close wake passages (within 3 ship widths), the single wakes, the double wakes and for all wakes in the datset.*

| | **Bubble wake depth [m]** | | | | | **Bubble wake longevity [min]** | | | | | |
|---|---|---|---|---|---|---|---|---|---|---|---|
| | Mean | Median | Q25 | Q75 | Std | Mean | Median | Q25 | Q75 | Std | **n** |
| **Close wakes** | 11.8 | 11.5 | 9.5 | 13.5 | 4.3 | 00:11:00 | 00:09:59 | 00:06:29 | 00:13:15 | 00:06:34 | 39 |
| **All wakes** | 10.3 | 9.5 | 7.5 | 12.5 | 4.1 | 00:10:14 | 00:08:00 | 00:05:29 | 00:13:29 | 00:06:29 | 69 |
| **All double** | 11.2 | 10.5 | 8.5 | 13.5 | 4.4 | 00:12:21 | 00:11:29 | 00:07:00 | 00:19:00 | 00:06:23 | 27 |
| **All** | 10.6 | 9.5 | 7.5 | 12.5 | 4.2 | 00:10:50 | 00:08:44 | 00:05:53 | 00:15:45 | 00:06:29 | 96 |
| | **$\varepsilon$ wake depth [m]** | | | | | **$\varepsilon$ wake longevity [min]** | | | | | |
| | Mean | Median | Q25 | Q75 | Std | Mean | Median | Q25 | Q75 | Std | **n** |
| **Close wakes** | 13.4 | 13.5 | 11.5 | 14.5 | 3.7 | 00:06:17 | 00:05:59 | 00:04:45 | 00:07:44 | 00:02:33 | 39 |
| **All wakes** | 11.8 | 11.5 | 9.5 | 13.5 | 3.9 | 00:06:22 | 00:05:59 | 00:04:59 | 00:07:59 | 00:02:41 | 69 |
| **All double** | 12.9 | 11.5 | 9.5 | 17.0 | 3.8 | 00:09:07 | 00:08:00 | 00:06:44 | 00:10:14 | 00:03:53 | 27 |
| **All** | 12.1 | 11.5 | 9.5 | 14.5 | 3.9 | 00:07:08 | 00:06:30 | 00:05:00 | 00:08:30 | 00:03:18 | 96 |

| | **Distance to instrument [m]** | | | | | |
|---|---|---|---|---|---|---|
| | Mean | Median | Q25 | Q75 | Std | **n** |
| **Close wakes** | 32 | 29 | 16 | 42 | 21 | 39 |
| **All wakes** | 64 | 46 | 26 | 101 | 51 | 69 |
| **All double** | 31 | 18 | 9 | 46 | 32 | 27 |
| **All** | 55 | 38 | 16 | 82 | 49 | 96 |

The change of figures and tables will naturally be accompanied with a revised description and analysis of the result. The main findings, the statistics for the entire dataset will still be the same, thus the main findings will not change. However, the statistics for the closest passages will have slightly deeper and longer wakes, compared to the entire dataset (Table 1). As we consider this part of the dataset more representative, this will only strengthen the overall argument that the temporal and spatial scales of ship wakes are large enough to take into account in areas with intense ship traffic. We also suggest using the median values from the close passage category for the calculations in the example in section 3.3 in the manuscript.

Regarding the general comment "*The main finding of this study is that turbulent ship wakes can reach deeper than the previously observed values, no physical explanation or discussion is provided*". In relation to the new figures in the revised result section, we will develop the discussion regarding possible physical explanations to the variation in wake depth and longevity. However, there are two main reason to why this was not extensively discussed in the submitted manuscript.

Firstly, the aim of the study is to describe the spatiotemporal characteristics of the turbulent ship wake, and not to fully explain the physical variation. Nevertheless, we fully agree with and acknowledge the potential of further analysis of this phenomenon. However, resolving all the parameters determining the characteristics and impact of ship-induced turbulent wakes, will require years of further studies. Therefore, there is a need for a first, more descriptive study, which will provide an understanding of the relevant scales and parameters to consider in future studies, as well

as a well described methodology. However, we acknowledge that the aim can be made clearer, and we therefore suggest changing the title to: "*In situ* observations of turbulent ship wakes and their spatiotemporal extent". In addition, we will rephrase the last paragraph (lines 105–115) in the introduction, to further motivate and describe the aim of the paper.

Secondly, we did look at possible physical explanations to the variation in wake depth and longevity, but due to low explanatory power, these results were only briefly mentioned, but not included in the submitted manuscript. We agree that there is a need to expand this discussion in the manuscript, and we will do so by using the vessel-related data we have. If requested, we could also include figures relating wake depth and longevity to ship length, draught, width, and speed, (Figure 6), as a basis for this discussion. However, in that case we propose including them in the supplementary information.

[Figure]

*Figure 6. Example figures of how the vessel length, draught, width and speed relates to the ε maximum wake depth and bubble wake longevity, for all single passages in the dataset. Note that for all parameters but vessel speed, the categories with the highest values (rightmost bars) have very few passages (<10).*

*It is not clear why the authors chose two separate locations for the in-situ and remote sensing study. Is it possible to find satellite imagery for the in-situ measurement period?*

We appreciate this very relevant comment. There is of course satellite imagery available for the *in-situ* study area as well, and we have retrieved it for the *in-situ* study period. However, the satellite passes only every 16th day, and during the study period there were no cloud-free images.

Regarding why we chose two different study areas all together, it was due to logistical reasons. The Bornholm study area was chosen as the ideal spot for the satellite study, as it one of the most intensely trafficked areas in the Baltic Sea. Initially, instruments were placed in the satellite study area, but due to unfortunate events they were lost. To have a better possibility to monitor the instrument, but still have an area with a lot of ship passages, we chose the Gothenburg harbour area for the second attempt of an *in-situ* study. The reason we still chose to keep the Bornholm are for the satellite study, was due to the more favourable weather conditions. A cloud free sky is needed for the satellite images to be usable, and since it only passes two times per month, the rainy

Gothenburg area was ill suited for the satellite study. We did not find this information suitable to include in the manuscript, but if requested we will motivate the choice of study area in more detail.

We suggest adding a comment in the manuscript, explaining the lack of cloud free satellite images from the *in-situ* study area and period, and the motivation for choosing the two different sites.

*Also, vertical profiles should have been measured more frequently to see the effect of wakes on stratification and mixing.*

We agree and acknowledge this as the main potential improvement of our observations, which we have also discussed in line 606-618. Furthermore, the long-term aim of our research is to be able to study and discuss the effect of wakes on stratification and mixing. However, the aim of the current study was to describe the spatiotemporal scales of the turbulent wake, and not resolving all the parameters determining the effect of wakes on stratification and mixing. As mention in a previous answer, we realise that the current title and aim, could give the impression that the paper is of a more explanatory nature. However, we consider the current amount of vertical profiles enough for the aim of this paper and suggest leaving the discussion regarding the effect of wakes on stratification and mixing for the next paper (where high-resolution profiling will be included).

To address this comment, we will clarify the aim of the paper as being mostly descriptive.

**Specific comments:**

*Line 191: Capitalize Python*

To address this comment, we will Capitalize Python on line 191.

*Line 224: Why 15%? Please justify.*

As stated in line 225-226, the wake area was manually defined using imagery of the echo amplitude and dissipation rate of turbulent kinetic energy. The exact value used for delimiting the wake region, was manually adjusted for each wake, using trial and error until the defined area sufficiently overlapped with the wake region in the image. Hence, the limit was chosen as the approximate value were noise was excluded but the wake region was included. However, this value was not decided upon in advance, but rather after the analysis it was clear that most of the values were approximately 15% higher compared to the daily/nightly mean.

To address this comment, we suggest clarifying that this value was not chosen in advance but was manually adjusted based on visual scrutiny of plotted figures, but that most values were ~15% higher than the daily/nightly mean.

*Line 355-359: Why do the bubble wakes look different from turbulence? Please discuss further.*

We acknowledge that this can be discussed further. The bubbles are an indication that surface water is mixed down at depth and has been mixed with the ambient water. They will remain there, or rise or collapse with time depending on size etc. The dissipation, on the other hand, is a measure of the turbulent motions in the water that mixes the water down. When the turbulence decays (due to dissipation) the dissipation also decays and dies out. But the bubbles may remain after that has happened.

The dissipation estimate is also influenced by neighbouring cells (equation 1), so the estimate may be deeper just due to the method used. I.e. if there is strong turbulence in one cell and none in the next, the method may still show some turbulence in the calm cell.

To address this comment, we suggest expanding the discussion regarding why the bubble wakes look different from turbulence, including the effect of bubble rise, collapse and retention, as well as possible biases due to the method used to estimate the dissipation.

*Line 550-552: This seems to be a negative result: the stratification was not affected by the wake. Remove this part. I suggest that the authors measure more vertical profiles in the study area and/or provide literature for more data to characterize general and unusual environmental conditions. 4 casts x 2 days are not enough.*

As mentioned in a previous comment, we agree and acknowledge that the amount of vertical profiles is too low to have an in-depth discussion regarding the effects of ship-induced vertical mixing on stratification. However, for the aim of this paper, we consider the current amount of profiles enough to describe the spatiotemporal scales of the turbulent wake.

However, regarding the comment related to the sentence above, we humbly disagree that it is a negative result. Turbulence will not be able to reach below the mixed layer, to 17.5 m depth, without mixing the water. Thus, we find it highly unlikely that the thermal stratification at 5 m was present within the wake. This means that the stratification was influenced by the wake and that waters above and below the thermocline were mixed with each other. However, three hours later the water has re-stratified and the mixed water has spread out laterally. We do not claim that there was a long-lasting effect on the stratification. However, during the longevity of the wake the stratification was most likely affected. Hence, in a scenario with very frequent ship passages, there will be less time for the re-stratification to occur, and a more long-lasting effect on the stratification could be possible.

To address this comment, we will clarify the aim of the paper as being mostly descriptive. However, we suggest keeping the current sentence, based on the arguments provided above.

*Line 574-581: As mentioned above, please try to find satellite imagery that covers the in-situ measurement area.*

As mentioned in a previous answer, satellite imagery covering the in-situ measurement area and period has been retrieved, but it was covered by clouds and impossible to use. A cloud free sky is needed for the satellite images to be usable, and the *in-situ* measurement area is too rainy to have a sufficient amount of cloud free images for the satellite study. Moreover, the Gothenburg area does not have as many ship passages as the Bornholm area, which is essential as you need to be sure to have ship passages at the time of the satellite passage. See also the previous answer regarding the choice of study area for the satellite study.

To address this comment, we suggest adding a sentence in the manuscript explaining the lack of cloud free satellite images from the *in-situ* study area and period, and the motivation for choosing the two different sites.

*Line 587: Note that winds are also important.*

We agree that winds are important, both as they affect waves and currents. As we have measurements of the water speed and waves, we consider the wind effect on currents and waves to

be captured by those measurements. However, we acknowledge that the wind should still be mentioned.

To address this comment, we suggest adding wind to the sentence in line 587 and stating that as we have measured the water speed and waves, we consider the wind effect to be captured by those measurements.

***Line 618:*** *What parameters?*

To address this comment, we will name the parameters that have been discussed in the paper in relation to the ship wake depth, to make it clear which parameters we mean.

---

## Author Response (AR1)

| Nr. | Review comment | Author's answer | Change in manuscript |
|---|---|---|---|
| 1 | The paper is mostly descriptive and practically no physical analysis of the observations is performed. It looks mostly like a report on some routine observations, like "a pile of data"… | As indicated by the title, we acknowledge that the aim of the paper is to describe the phenomenon of turbulent wakes, and thus being *"mostly descriptive"*. However, we humbly disagree that the paper is *"mostly like a report on some routine observations"* and that *"practically no physical analysis of the observations is performed"*. Even though the aim of the paper is to describe the extent of wake influence, we have made analysis of the turbulence in the wakes (dissipation rate of turbulent kinetic energy). Furthermore, we have presented an example of what the described spatiotemporal extent of the wake would implicate in terms of temporal and spatial wake-impact in a highly frequented ship lane. Nevertheless, we fully agree with and acknowledge the potential of further analysis of this phenomenon. However, resolving all the parameters determining the characteristics and impact of ship-induced turbulent wakes, will require years of further studies. Therefore, there is a need for a first, more descriptive study, which will provide an understanding of the relevant scales and parameters to consider in future studies, as well as a well described methodology. The work of measuring large quantities of turbulent wakes in ship lanes, is far from routine, and not many observations have so far been published. In addition, we would like to highlight the interdisciplinary nature of this work. The process of identifying and linking the wakes with the ship inducing the wake is time-consuming, but necessary to relate the wakes to the physical properties of the vessels. However, we agree and acknowledge that the vessel-related analysis can be developed further, and we have suggested how to do so in detail in the answer to that specific comment below. | Changed title, revised paragraph to clarify the aim and further motivate the need for the study. |
| 2 | …only methodological aspects of the work are discribed, although not clearly enough (see, some specific comments below) | The specific comments are addressed individually below. Furthermore, we agree and acknowledge that there is a focus on describing the methodology of the work. That is because there are currently no published descriptions of best practice or standard methodologies to study ship wakes and passages in heavily trafficked ship lanes during an extended period of time without interfering with the traffic itself. It is an activity with many technical and practical challenges, and therefore, the methodology has been explained in detail. Moreover, most previous studies of the turbulent ship wake, have been performed using echo sounders/multibeam mounted on the ship hull of a small ship, which have travelled across the ship wake in serpentine movements behind the ship, measuring the wake from above. Here we propose a method based on upward facing instruments placed on the sea floor under the ship lane. We therefore consider it important to describe and discuss the methodology in detail, and we propose to add additional figures to illustrate the experimental setup, to clarify the questions asked. | For changes in manuscript see answer to the specific comments below. |

| 3 | "One cannot find in the text any new physical effects" and "The paper in its present form does not look interesting and informative from a scientific point of view" | Regarding the general comment *"One cannot find in the text any new physical effects"* and *"The paper in its present form does not look interesting and informative from a scientific point of view"*, we respectfully disagree. To our knowledge, there is no published dataset showing this consistent pattern of turbulent and temporal wakes, and no studies including more than a handful of different ships. The observed maximum wake depth presented in this study, exceed previously reported wake depths, which sheds new light on a novel perspective of the environmental impact from shipping. Moreover, the longevity and persistence of the thermal wakes shown in the satellite data highlights the importance of considering the impact of ship wakes in highly trafficked areas (with up 55 000 passages per year). This effect is important to raise awareness of, especially within the FerryBox community, as the temperature measurements made within ship lanes can be biased if they are made in the wake of another ship. The longevity of the temperature difference indicates that the ship wake water stays a separate entity (is not diluted/mixed) for a substantial period of time (+ 1 hour). Even though only temperature has been measured in this study, the thermal signal can be considered as proxy for the ship wake water, and potential changes in other chemical/physical parameters such as salinity, nutrient concentration etc. should also be sustained in the ship wake water as long as it is not mixed with the surrounding water mass. | This comment is addressed by the revision of the manuscript which clarifies the aim and motivation of the paper. See comment 9 and 1. |
| 4 | The aim/motivation of the paper is not clear | Firstly, we suggest changing the title to: "*In situ* observations of turbulent ship wakes and their spatiotemporal extent", to clarify that the aim is to describe the characteristics of the turbulent ship wake. We will also rephrase the last paragraph (lines 105–115) in the introduction as below, to further motivate and describe the aim of the paper: | The title has been changed to clarify that the aim of the paper is to describe the spatiotemporal extent of turbulent ship wakes. |
| 5 | An error in formula (1) | We will correct the error. | The error has been amer |

| 6 | A scheme of the ADCP deployment and recording of ship wakes has to be presented to understand how the ship wakes are recorded by the ADCP. For instance, the bubble wake manifestations similar to one in Fig.2 appear when the ADCP is towed across the ship wake, or if the wake is moving in the cross wake direction due to currents passing by the ADCP beams. How thus the record of a ship wake in Fig.2 could be obtained for a stationary looking upward ADCP? Was that due to a current moving a wake through a zone illuminated by ADCP? | We appreciate the suggestion of adding as scheme to describe the instrument deployment and recording. We suggest adding a figure describing the instrumental setup in the material and methods section and a more detailed description of what is measured and how. Moreover, we suggest adding an additional sketch in Figure 2 in the manuscript, to further illustrate what the ADCP is recording. In addition, we have also made some complimentary illustrations that could be added to the manuscript or a supplementary information section, if requested. | A sentence has been added to clarify that the ADCP measures the wake development thorough time, with a reference to figure 3. A figure of the experimental setup has been added. Previous figure 2 (now 3) has been revised to include an indication of the ship passage to clarify what the ADCP instrument is measuring. The possible reasons to why the wake signal is detected from ships passing up to 184 m from the ADCP instrument is discussed in detail in the answer to question 7. |

| 7 | Why the wakes appeared in the ADCP records for ships passed by at some distances from the ADCP? Because of the wake turbulent diffusion? If so, why not to analyze, e.g. the characteristic times of the turbulent diffusion, the diffusion spatial/temporal decay, etc.? | We consider two main reasons for the wakes appearing in the ADCP records for ships passing at some distance from the wake. Firstly, currents can move the wake towards the instrument. However, when looking at the impact of currents in our data, we could not see that that current speed and/or direction was affecting the existence of a wake in our data. Moreover, during most of the measurement period the current speed at the instrument position was very low. These two circumstances indicate that wake drift due to currents is probably not the main reason that wakes are detected from ships passing at long distances from the instrument. The second possible reason, as suggested, is that turbulent diffusion widens the wake. As shown in the satellite data, the median thermal wake width was 157 mindicationg, that a ship passage at 150–200 m away from the instrument could induce a wake, where the edge of the wake eventually reaches the instrument. In addition, the wake will be widened by buoyancy effects if the water is stratified. There is also a reason to why we have not analysed the characteristic times of the turbulent diffusion. As the ADCP only measure in one point, it is not clear if the entire length of the wake is captured or which part of the wake is being measured. Since it is not sure that the entire wake from start to end is measured, making calculations on diffusion rate based on it would not be fully representative. For the aim of the current paper – estimating the temporals extent of the turbulent wake, we considered the large quantity of measurements sufficient to give an estimate of the overall/general temporal extent, without analysing the characteristic times of the turbulent diffusion. | A sentence has been added to explain why wakes from ships passing at some distances from the ADCP can be detected. |
| 8 | Line 301. I cannot understand how this can happen : "…when two ships passed the instrument at the same time" | The reason why two ships can "pass at the same time" are two. Firstly, large ships may require pilot assistance and/or tugboats to enter the harbour. In these cases, the ships pass right next to each other and it is impossible to separate the wakes form the different ships (see figure xx in supplementary info). Most of the double passages in the dataset where these types of occasions. The other occasion of double passages is related to the width of the wake and the fact that the wake from ships can be detected even when ships are passing at a distance from the instrument. There were a few occasions where two ships passed the instrument at the same distance from the instrument, but on different sides (figure xx in supplementary info). In these cases, it was not possible to tell which of the ships that induced the wake detected by the ADCP, as our analysis of the data showed now clear correlation between current direction and wake detectability. Hence wakes from these occasions were treated as a double passage, as the detected wake could come from either ship or be a combination of both passages. We will add a sentece clarifying in which occasions two ships can "pass at the same time". | A clarification to how to ships can pass the instrument at the same time has been added. |

| | | | |
|---|---|---|---|
| 9 | Categorization of the ships in the context of their turbulent wakes does not look physically justified. More reasonable would be to relate the wakes to the ship weight, draught, speed, possibly to the size/number of propellers | We acknowledge the comment to relate the wake depth and longevity to another parameter than ship type, and therefore suggest a revision of the result section. It is beyond the scope of this paper to investigate the dependence between various non-dimensional parameters, which would be the most physically justified thing to do. However, we expect that the wake size to a large degree depends on the force or power put into the water by the propeller. We therefore propose replacing the current figures 5 and 6, to figures relating the wake depth and longevity to force (F), calculated as $\rho$*ship width*ship draught*ship speed^2 [kg m s$^{-2}$], with seawater density ($\rho$) equal to 1025 kg m$^{-3}$. This parameter is proportional to ship drag and will relate the wake depth and longevity to vessel size and speed, which we agree are parameters expected to have an impact on the formation of the turbulent wake. In addition to the change in figures, we also propose a change to table 2 and 3 in the manuscript. We will remove the ship type category statistics and instead include statistics for the close passage category (0-3 ship widths). The change of figures and tables will naturally be accompanied with a revised description and analysis of the result. | The methods section has been revised to describe the new presentation of the data, removing the ship type categories and adding the calculation of ship Force. The result and discussion section has been revised to describe the new presentation of the data, removing the ship type categories and adding the calculation of ship Force. This provides a basis for discussing possible physical explanations to the variation in wake depth and longevity in the detected wakes, as well as the importance of ship size and speed. |
| 10 | line 331 "As the fraction of detected induced wakes at similar distances differ between ship types, it is an indication that the ship type impacts the characteristic of the turbulent wake" . I disagree with the statement and I think that the difference is determined mostly by the ship weigh and ship speed. | This comment is addressed in detail in the answer to comment 5. As we now suggest replacing the figures which presents the data based on ship type, this sentence will be removed. | This comment is addressed in detail in the answer to comment 9. As we now suggest replacing the figures which presents the data based on ship type, this sentence will be removed. |

| 11 | The paper is full of obvious, trivial statements, e.g. "in general the deepest wakes were caused by ships passing closer to the instrument, whereas ships passing at larger distances from the instrument (100–199 m) mainly caused shallower wakes : : :" (lines 369-370) "the maximum dissipation rates : : : in the core of the wake : : :.are : : :..much larger than what is usually observed in the core of, or below, the surface mixed layer" (lines 403-405), etc. etc. | We acknowledge and understand there are statements in the manuscript that can be perceived as trivial and obvious, depending on the researcher's specialisation. To balance the content to suit a diverse audience, from different highly specialised disciplines, is a general challenge in interdisciplinary research. Therefore, the second example in this comment was included for a reason. Our aim is to reach an interdisciplinary audience within ocean science, in line with the scope of this journal. This specific comment was included because the non-oceanographic co-authors of the paper explicitly asked for a comparison between our measured values and values that would occur naturally in the system. We believe that these types of statements fill an important function in making the content of the paper more accessible to an interdisciplinary audience. However, with the new suggested figures, much of the result section will be rewritten, and will make sure not to pay extra attention to this aspect. | With the revised result section, the specified sentences have been removed. |
| --- | --- | --- | --- |
| 12 | The main finding of this study is that turbulent ship wakes can reach deeper than the previously observed values, no physical explanation or discussion is provided. | In relation to the new figures in the revised result section, we will develop the discussion regarding possible physical explanations to the variation in wake depth and longevity. See answer to commment 1 and 9. | See answers and revisions to Comment 1 and 9. |
| 13 | It is expected that ship specifications and speeds determine ship wake depths, so the authors should be able to discuss further based on the available ship information. | We acknowledge the comment to relate the wake depth and longevity to another parameter than ship type, and therefore suggest a revision of the result section. Se answer to reviewer comment 9. | |

| 14 | It is not clear why the authors chose two separate locations for the in-situ and remote sensing study. Is it possible to find satellite imagery for the in-situ measurement period? | We appreciate this very relevant comment. There is of course satellite imagery available for the in-situ study area as well, and we have retrieved it for the in-situ study period. However, the satellite passes only every 16th day, and during the study period there were no cloud-free images. Regarding why we chose two different study areas all together, it was due to logistical reasons. The Bornholm study area was chosen as the ideal spot for the satellite study, as it one of the most intensely trafficked areas in the Baltic Sea. Initially, instruments were placed in the satellite study area, but due to unfortunate events they were lost. To have a better possibility to monitor the instrument, but still have an area with a lot of ship passages, we chose the Gothenburg harbour area for the second attempt of an in-situ study. The reason we still chose to keep the Bornholm are for the satellite study, was due to the more favourable weather conditions. A cloud free sky is needed for the satellite images to be usable, and since it only passes two times per month, the rainy Gothenburg area was ill suited for the satellite study. We did not find this information suitable to include in the manuscript, but if requested we will motivate the choice of study area in more detail. | A motivation to why the Bornholm study area was chosen for the satellite analysis, instead of the Gothenburg study site, has been added in the method and discussion section. |
| --- | --- | --- | --- |
| 15 | Also, vertical profiles should have been measured more frequently to see the effect of wakes on stratification and mixing. | We agree and acknowledge this as the main potential improvement of our observations, which we have also discussed in this section of the manuscript. Furthermore, the long-term aim of our research is to be able to study and discuss the effect of wakes on stratification and mixing. However, the aim of the current study was to describe the spatiotemporal scales of the turbulent wake, and not resolving all the parameters determining the effect of wakes on stratification and mixing. As mention in a previous answer, we realise that the current title and aim, could give the impression that the paper is of a more explanatory nature. However, we consider the current amount of vertical profiles enough for the aim of this paper and suggest leaving the discussion regarding the effect of wakes on stratification and mixing for the next paper (where high-resolution profiling will be included). | To address this comment, we will clarify the aim of the paper as being mostly descriptive. |
| 16 | Line 191: Capitalize Python | To address this comment, we will Capitalize Python on line 191. | Python has been capitalised. |
| 17 | Line 224: Why 15%? Please justify. | As stated, the wake area was manually defined using imagery of the echo amplitude and dissipation rate of turbulent kinetic energy. The exact value used for delimiting the wake region, was manually adjusted for each wake, using trial and error until the defined area sufficiently overlapped with the wake region in the image. Hence, the limit was chosen as the approximate value were noise was excluded but the wake region was included. However, this value was not decided upon in advance, but rather after the analysis it was clear that most of the values were approximately 15% higher compared to the daily/nightly mean. | It has been clarified that the value of 15 % was not chosen in advance but was manually adjusted based on visual scrutiny of plotted figures, but that most values were ~15% higher than the daily/nightly mean. |

| | | |
|---|---|---|
| 18 Line 355-359: Why do the bubble wakes look different from turbulence? Please discuss further. | We acknowledge that this can be discussed further. The bubbles are an indication that surface water is mixed down at depth and has been mixed with the ambient water. They will remain there, or rise or collapse with time depending on size etc. The dissipation, on the other hand, is a measure of the turbulent motions in the water that mixes the water down. When the turbulence decays (due to dissipation) the dissipation also decays and dies out. But the bubbles may remain after that has happened. The dissipation estimate is also influenced by neighbouring cells (equation 1), so the estimate may be deeper just due to the method used. I.e. if there is strong turbulence in one cell and none in the next, the method may still show some turbulence in the calm cell. | The discussion regarding the differences between the bubble and turbulent wake has been expanded. |
| 19 Line 550-552: This seems to be a negative result: the stratification was not affected by the wake. Remove this part. I suggest that the authors measure more vertical profiles in the study area and/or provide literature for more data to characterize general and unusual environmental conditions. 4 casts x 2 days are not enough. | See answer to comment 15 regarding measuring more vertical profiles. However, regarding the specfic section mentioned here, we humbly disagree that it is a negative result. Turbulence will not be able to reach below the mixed layer, to 17.5 m depth, without mixing the water. Thus, we find it highly unlikely that the thermal stratification at 5 m was present within the wake. This means that the stratification was influenced by the wake and that waters above and below the thermocline were mixed with each other. However, three hours later the water has re-stratified and the mixed water has spread out laterally. We do not claim that there was a long-lasting effect on the stratification. However, during the longevity of the wake the stratification was most likely affected. Hence, in a scenario with very frequent ship passages, there will be less time for the re-stratification to occur, and a more long-lasting effect on the stratification could be possible. | The example of the ship passage affecting the thermal stratification has been clarified. For the comment regarding measuring more vertical profiles in the study area, see Comment 15. |
| 20 Line 574-581: As mentioned above, please try to find satellite imagery that covers the in-situ measurement area. | See answer to comment 14. | A section has been added in the discussion and method, motivating the choice of the two different sites. A sentence has been added in the discussion suggesting future improvements of the method. See answer to Comment 14. |
| 21 Line 587: Note that winds are also important. | We agree that winds are important, both as they affect waves and currents. As we have measurements of the water speed and waves, we consider the wind effect on currents and waves to be captured by those measurements. However, we acknowledge that the wind should still be mentioned. | Wind is now also mentioned. As we have measured the water speed and waves, we consider the wind effect to be captured by those measurements. |
| 22 Line 618: What parameters? | To address this comment, we will name the parameters that have been discussed in the paper in relation to the ship wake depth, to make it clear which parameters we mean. | The parameters have been specified. |

**In situ* observations of turbulent ship wakes and their spatiotemporal extent**

Amanda T. Nylund[1], Lars Arneborg[2], Anders Tengberg[1], Ulf Mallast[3], Ida-Maja Hassellöv[1]

[1]Department of Mechanics and Maritime Sciences, Chalmers University of Technology, Gothenburg, 412 96 Gothenburg, Sweden.
[2]Swedish Meteorological and Hydrological Institute (SMHI), Gothenburg, 426 71 Västra Frölunda, Sweden.
[3]Department Monitoring and Exploration Technologies, Helmholtz Centre for Environmental Research, Leipzig, 04318 Leipzig, Germany.

*Correspondence to*: Ida-Maja Hassellöv (ida-maja@chalmers.se)

**Abstract.** In areas of intensive ship traffic, ships pass every ten minutes. Considering the amount of ship traffic and the  predicted  increase in global maritime trade, there is a need to consider all type of impacts shipping has on the marine environment While there is increasing awareness about, and efforts to reduce, chemical pollution from ships, less in known about  physical disturbances and ship-induced turbulence has so far been completely neglected. To address the potential importance of ship-induced turbulence on e.g. gas exchange, dispersion of pollutants, and biogeochemical processes, there is an immediate need for characterisation of the temporal and spatial scales of the turbulent wake.  There is a lack of field measurements of turbulent wakes of real-size ships, and this study addresses that gap by *in situ* and *ex situ* measurements of the depth, width, length, intensity and longevity of the turbulent wake for ∼240 ship passages of differently sized ships. A bottom-mounted Acoustic Doppler Current Profiler (ADCP) was placed at 32 m depth below the ship lane outside Gothenburg harbour, and used to measure wake depth and temporal longevity. Thermal satellite images of the Thermal Infrared Sensor (TIRS) onboard Landsat 8 were used to measure thermal wake width and spatial longevity, using satellite scenes from the major ship lane North of Bornholm, Baltic Sea. Automatic Information System (AIS) records from both the investigated areas were used to identify the ships inducing the wakes. The results from the ADCP measurements show median wake depths of ∼10 m, and several occasions of wakes reaching depths > 18 m. The temporal longevity of the wakes had a median of around 8 min and several passages of > 20 min. The satellite analysis showed a median thermal wake length of 13.7 km, and the longest wake extended over 60 km, which would correspond to a temporal longevity of 1 h 42 min (for a ship speed of 20 knots). The median thermal wake width was 157.5 m. The measurements of the spatial and temporal scales are in line with previous studies, but the deep mixing and extensive longevity presented in this study, has not previously been documented. The results from this study show that ship-induced turbulence occurs at temporal and spatial scales large enough to imply

**Commented [AN1]:** **Comment 4** (R1, general comment): The aim/motivation of the paper is not clear.

**Commented [AN2R1]:** Changes:
- The title has been changed to clarify that the aim of the paper to describe the spatiotemporal extent of turbulent ship wakes.

[revised manuscript text omitted]

**Commented [AN3]: Comment 15** (R2, general comment): Also, vertical profiles should have been measured more frequently see the effect of wakes on stratification and mixing.

**Commented [AN4R3]:** Changes:
- The aim of the paper is to describe the spatiotemporal extent the wake, and not to investigate the effect on stratification and mixing. We acknowledge that the previous title and introduction might have given the impression that we aimed to discuss the effect on stratification and mixing. Therefore, we have address this comment by clarifying the aim of the paper by revising the title and the last paragraph of the introduction.

**Commented [AN5]: Comment 14** (R2, general comment): It is not clear why the authors chose two separate locations for the situ and remote sensing study. Is it possible to find satellite image for the in-situ measurement period?

**Commented [AN6R5]:** Changes:
- A motivation to why the Gothenburg study site was chosen in favour of the Bornholm study area, has been added.

[Figure]

**Figure 1: Overview of the two study areas (a) showing the location of the ADCP under the ship lane outside Gothenburg (b) and the area covered by the analysed satellite images (c). White dashed lines indicate ship routes of ferry lines and the boxes in (c) indicate the area defined as the ship lane area in the satellite image analysis. The ship lane and traffic separated zone north of Bornholm are shown in Figure 12.**

**2.1.1 Field measurements and data collection**

A bottom-mounted Nortek Signature 500 kHz broadband Acoustic Doppler Current Profiler (ADCP) was deployed under the ship lane (57.61178 N, 11.66102 E), fixed in upward-looking position in a bottom frame (Figure 2). Similar setups have previously been used to study the bubble cloud of the turbulent wake, by Trevorrow et al. (1994) and Weber et al. (2005). The instrumental setup provides measurements of the overlaying water column trough time (Figure 3), hence, recording the wake development in a fixed point through time. Under the assumption of a stationary wake moving with the ship velocity, the observations can also be interpreted in terms of the spatial change of the wake with distance from the ship. The instrument was deployed at approximately 30 m depth, for a duration of 4 weeks (28 August to 25 September 2018). The ADCP measured along beam current velocities, using four slanted beams (25° angle) and one vertical beam (ping frequency 1 Hz, cell size 1 m on all beams). The echo amplitudes from the beams were also used to detect the wake bubbles. All single ping data on currents and echo amplitude was stored on-board the instruments and analysed, see sect. 2.1.2. The range of sonar frequencies that are suitable for detecting bubbles in the turbulent ship wake is 30 kHz to 1 MHz and depends on the size of the bubbles in the wake (Liefvendahl and Wikström, 2016). A SonTek CastAway®-CTD (Xylem, San Diego, California) was used to measure salinity and temperature profiles at the time of the instrument deployment (August 28, 2018, 4 casts) and retrieval (September 25, 2018, 4 casts).

**Commented [AN7]:** Comment 6 (R1, specific comment 2): A scheme of the ADCP deployment and recording of ship wakes is to be presented to understand how the ship wakes are recorded by ADCP. For instance, the bubble wake manifestations similar to one in Fig.2 appear when the ADCP is towed across the ship wake, or if the wake is moving in the cross wake direction due to currents passing the ADCP beams.

(this part of the question is addressed further down)
How thus the record of a ship wake in Fig.2 could be obtained for a stationary looking upward ADCP? Was that due to a current moving a wake through a zone illuminated by ADCP?

**Commented [AN8R7]:** Changes:
-A figure illustrating the deployment setup has been added.
- A sentence has been added to clarify that the ADCP measures the wake development thorough time, with a reference to figure 3.

**Commented [AN9]:** Comment 15 (R2, general comment): Also, vertical profiles should have been measured more frequently to see the effect of wakes on stratification and mixing.

**Commented [AN10R9]:** To address this comment, we will clarify the aim of the paper as being mostly descriptive. See introduction and discussion for changes.

**Commented [AN11R9]:** We agree and acknowledge this as the main potential improvement of our observations, which we have also discussed in section 3.4 Limitations and Future outlook. Furthermore, the long-term aim of our research is to be able to study and discuss the effect of wakes on stratification and mixing. However, the aim of the current study was to describe the spatiotemporal extent of the turbulent wake, and not resolving all the parameters determining the effect of wakes on stratification and mixing. We realise that the current title and aim, could give the impression that the paper is of a more explanatory nature. However, we consider the current amount of vertical profiles enough for the aim of this paper and suggest leaving the discussion regarding the effect of wakes on stratification and mixing for the next paper (where high-resolution profiling will be included).

175

[Figure]

**Figure 2:** Scheme of instrument deployment, showing a) the sideview with perspective of the ADCP, placed on the seafloor facing upward, and recording the turbulent wakes during  ship passages, and b) a top view perspective of the ADCP recording bubbles from a turbulent wake, induced by a ship passing above, but slightly to the side of the instrument.

180

A dataset of the ships passing the study area during the field measurement period, was purchased from the Swedish Maritime Administration. The dataset is from the Baltic Marine Environment Protection Commission (HELCOM) Automatic Information System (AIS) database, which is processed according to the procedure described in the annex of the HELCOM Assessment on maritime activities in the Baltic Sea 2018 (HELCOM, 2018). The Swedish Institute for the Marine Environment (SIME) provided additional files from the same HELCOM database, with AIS data for the analysed satellite scenes and the Gothenburg harbour study area. Vessel information from MarineTraffic – Global Ship Tracking Intelligence (www.marinetraffic.com) was used to retrieve detailed information about the width, length and draught of the ships in the dataset.

**2.1.2 Data analysis**

**Compiling the ADCP wake dataset**

All ship wakes in the dataset were identified manually using high resolution figures of the echo amplitude of the ADCP beams (see Fig. 3 for example). As the bubbles in the turbulent wake reflect the sound more efficiently than water, it results in an elevated echo amplitude in the turbulent wake region (NDRC, 1946; Marmorino and Trump, 1996; Trevorrow et al., 1994; Weber et al., 2005; Ermakov and Kapustin, 2010; Francisco et al., 2017). Generally, the wake signal could be clearly

distinguished from bubbles induced by waves or signal noise from fish or zooplankton. However, ambiguous cases were noted, and the wake dataset was therefore divided into wake categories based on the quality of the wake signal. Each wake in the dataset was then linked to a ship passing in the vicinity of the ADCP, using the HELCOM AIS dataset and manual comparison. This introduced additional uncertainties, as not all wakes had clear match with a ship passage. After incorporating the matching uncertainties, the final wake categories used in the analysis were: "**wake**", only including clear wakes with one clear match or delayed match; "**double**", clear wakes where two or three ships passed the instrument at the same time; and "**no wake**", which included all passages within 184 m of the instrument that did not induce a visible wake, as well as all uncertain wakes and matches, which were mostly due to windy conditions which created noisy data. The distance at which a wake can be detected from a passing ship is affected by wake broadening, drifting, and ship width. In this study, Tthe 184 m radius was chosen, as it was the furthest distance at which a clear wake and match was found in the dataset. There were two factors contributing to the existence of the "**double**" category. Firstly, the turbulent wakes in the dataset could be detected from ships passing at distances up to 184 m from the ADCP instrument. Hence, in cases when two ships passed at similar distances from the instrument at the time of a detected wake, it was not possible to distinguish which of the ships that induced the wake. Secondly, large ships may require pilot assistance and/or tugboats to enter the harbour. In these cases, the ships pass right next to each other and it is not possible to assign the wake to a single ship. In additionLastly, some wakes and passages were removed from the analysis altogether. These included ships with missing information in the AIS data (size information) and small sailing vessels, as they due to their small size and engine power were not deemed relevant for the investigated process.

**Commented [AN12]:** **Comment 8** (R1, specific comment 4). Line 301. I cannot understand how this can happen : "…when two ships passed the instrument at the same time"

**Commented [AN13R12]:** Change:
- The explanation of the passages in the double category has been expanded.

**Commented [AN14]:** **Comment 7** (R1, specific comment 3). Why the wakes appeared in the ADCP records for ships passed by some distances from the ADCP? Because of the wake turbulent diffusion? If so, why not to analyze, e.g. the characteristic times of the turbulent diffusion, the diffusion spatial/temporal decay, etc.?

**Commented [AN15R14]:** Changes:
- A sentence has been added to explain why wakes from ships passing at some distances from the ADCP can be detected.

(This comment is also addressed in the discussion section)

[Figure]

**Figure 3: Example of the bubble wake signal in the echo amplitude dataset (a) and the calculated dissipation rate of turbulent kinetic energy, $\varepsilon$, (b) from  the ADCP measurements. The upward facing ADCP was placed at approximately 30 m depth, repeatedly measuring the water column in one point. The dashed red line marks the time of ship passage. The high intensity red (a) and yellow (b) areas after the ship passage  the wake region. The increase of $\varepsilon$ down to the bottom is evidence of increased turbulence and a vertical mixing down to 30 m depth . The wake was induced by  a cargo ship  (width 25 m, length 229 m, draught 7 m), which passed the instrument at a distance of 34 m and a speed of 19 knots.**

**Distance calculation, AIS and ADCP dataset**

The AIS dataset included position reports for each ship every 2–10 seconds, which were used to calculate the ship's track. The closest distance between the ship-track and the vertical beam of the ADCP instrument was then calculated, using a local planar coordinate system, with the instrument at origo. The coordinates for the closest point on the track was also calculated, using the python GeoPy package function distance.distance, and the points just before and after the closest point on the track were then identified.

**Turbulence calculation, ADCP dataset**

The dissipation rate of turbulent kinetic energy is a measure for the strength of the turbulence. Per definition it is the rate of energy conversion from kinetic energy to heat due to viscous friction in the smallest eddies, but in a stratified water column it is also proportional to the mixing between different water masses. There are various ways of determining dissipation rates. In the present work it is estimated from the ADCP data using the structure function method (e.g. Lucas et al. (2014)), which estimates the dissipation rate of turbulent kinetic energy from the second-order structure function following Eq. (1):

**Commented [AN16]:** Comment 6 (R1, specific comment 2): Continuation…

How thus the record of a ship wake in Fig.2 could be obtained for stationary looking upward ADCP? Was that due to a current moving a wake through a zone illuminated by ADCP?

**Commented [AN17R16]:** Changes:
-A line indicating the time of the ship passage has been added to the figure to further relate the measurements to the ship wake.
- The possible reasons to why the wake signal is detected from ships passing up to 184 m from the ADCP instrument is discussed in detail in the answer to question 7.

**Commented [AN18]:** Comment 16 (R2, specific comment): Line 191: Capitalize Python

**Commented [AN19R18]:** Changes:
-Python is capitalised.

$$D_{11}(r, \Delta r) = \overline{\left( u_r{}'(r + \Delta r) - u_r{}'(r + \Delta r) \right)^2},$$

(1)

where $u_r{}'$ is the fluctuating velocity in the *r*-direction (in this case the beam direction), $\Delta r$ is the separation distance between two points along the beam, and overbar denotes time averaging. For separation distances shorter than the largest eddies the structure function relates to the dissipation rate and separation distance as in Eq. (2):

$$D_{11}(r, \Delta r) = C \varepsilon^{2/3} \Delta r^{2/3},$$

(2)

where $C$ is a universal constant. Since the shortest distance (the ADCP bin size) was 1 m, the method is only expected to work for very strong turbulence with vertical eddy scales of magnitude larger than 2–3 m.

For each ship wake in the "**wake**" and "**double**" category, the along beam current velocity measurements from the ADCP were used for turbulence calculations in the wake region. One of the slanting beams was malfunctioning but the four remaining beams were analysed. A 1-hour dataset following each passage, identified by the start of the bubble cloud, was analysed. Spikes deviating more than four times the standard deviation from the mean in overlapping windows of 100 sec length were removed. Since the velocity signal of surface waves at different depths may be expected to be coherent whereas turbulent signals are not, the two Empirical Orthogonal Function (EOF) modes with largest variance were removed from the series to reduce the influence of surface waves. A fourth order Butterworth high-pass filter with cut-off period 600 sec was used to extract the turbulent velocity fluctuations. The dissipation rate of turbulent kinetic energy was estimated in 30 sec bins using the structure function method according to the method described in Lucas et al. (2014). One dissipation rate estimate was based on the average of the result for the three slanting beams (see Fig. 2 for an example), and another was based on the vertical beam.

**Calculating wake depth, longevity, and maximum $\varepsilon$ intensity, ADCP dataset**

For each wake in the categories "**wake**" and "**double**", the wake region was defined for the parameters echo amplitude (bubble wake), dissipation rate of turbulent kinetic energy ($\varepsilon$), and the maximum velocity variance. To reduce noise in the dataset induced by turbidity at the sea floor, the data was normalised with respect to vertical distance from the instrument, assuming exponential decay of the signal strength. The wake region was defined by visual scrutiny of echo amplitude and $\varepsilon$ figures (see Fig. 2 for an example) and manually annotated. comparing the wake region to the daily/nightly mean, and all values ~15% higher than the mean was considered part of the wake. As this procedure often identified noise as part of the wake, The elevation in echo amplitude/$\varepsilon$ used for delimiting the wake region, as well as the depth and duration to consider, was manually adjusted for each wake to exclude noise. In general, the threshold was ~15% higher compared to the daily/nightly mean. both the percentage limit and the start time, stop time and maximum depth to include in the calculation, were manually defined for each wake to exclude noise. The deepest part of the wake region was used as a measure of the maximum wake depth and the maximum $\varepsilon$ intensity in the wake region was used as a measure of the maximum turbulence.

**Commented [AN20]:** **Comment 5** (R1, specific comment 1): An error in formula (1)

**Commented [AN21R20]:** Changes:
- The error has been amended.

**Commented [AN22]:** **Comment 17** (R2, specific comment): Line 224: Why 15%? Please justify.

**Commented [AN23R22]:** Changes:
- It has been clarified that the value of 15 % was not chosen in advance but was manually adjusted based on visual scrutiny of plotted figures, but that most values were ~15% higher than the daily/nightly mean.

The duration of the wake (temporal longevity in min) was calculated using the start time and end time of the wake region. All calculations were pursued using an individually developed Python code.

275

**Statistical analysis and graphical presentation of the ADCP wake dataset**

For the statistical analysis and graphical presentation, the categories "**wake**", "**double**", "**close wake**", and "**no wake**" were used. The category "**close wake**", comprises all wakes induced by ships passing within 0–3 ship widths from the instrument,
280 which roughly corresponds to 75 m. This cut-off was chosen as there was a substantial decrease in the percentage of induced wakes at passages > 3 ship widths from the instrument, indicating difficulties in detecting wakes at larger distances. As the wakes in the "**double**" category lack information about the ship inducing the wake, no "**double**" wakes are included in the "**close wake**" category,. hHence the double category is presented separately. The dataset was then analysed by ship type, using the five categories *cargo*, *tanker*, *passenger*, and the double categories *cargo + pilot* and *tanker + pilot*. For each ship
285 typecategory the median wake depth (m) and temporal wake longevity (min), was calculated for the bubble wake and the ε dissipation rate wake, together with standard deviation (std) and the 25th and 75th percentile. Furthermore, the percentage of ship passages that induced a visible wake in the ADCP beams was calculated along with the maximum ε intensity in the wake region. A Welch Analysis of Variance (ANOVA) was also performed, comparing the maximum wake depth and longevity between the five ship type categories.

290

For the graphical presentation, the wake depth and longevity results are presented in relation to vessel force (F) [kg m s$^{-2}$]. F was calculated from the ship width (B) [m], draught (T) [m], and speed (*s*) [m s$^{-1}$], as in Eq. (3):

$$F = \rho * B * T * s^2 \quad\quad\quad\quad\quad\quad\quad\quad\quad\quad (3)$$

with seawater density ($\rho$) equal to 1025 kg m$^{-3}$. The F parameter is proportional to ship drag and relates the wake depth and
295 longevity to vessel size and speed, which are parameters affecting the formation of the turbulent wake.

**2.2 Bornholm satellite study**

The Bornholm study area was chosen, as it covers the most intensely trafficked ship lane in the Baltic Sea, with approximately 50,000 ship passages per year (HELCOM, 2010). All large ships heading for the Eastern and Northern ports of the Baltic Sea, must use the Bornholm ship lane (HELCOM, 2018), which makes it ideal for studying ship-induced
300 vertical mixing from a variety of different ship types. Besides the purely traffic-related reason, a second reason for choosing the Bornholm area compared to the Gothenburg area, in which in.-situ data (ADCP, CTD) was retrieved was the availability of cloud-free satellite scenes, which is essential for detecting any surface object in the optical and thermal wavelength. The Bornholm area (path 193/ row 21) had 23 scenes with less than 23% cloud cover above the sea until August 2018, for the Gothenburg area (path 196/ row 20) only 9 scenes were available.

**Commented [AN24]:** **Comment 9** (R1, specific comment 5): Categorization of the ships in the context of their turbulent wakes does not look physically justified. More reasonable would be to r... the wakes to the ship weight, draught, speed, possibly to the size/number of propellers.

**Commented [AN25R24]:** Changes:
- The section has been revised to describe the new presentation... the data, removing the ship type categories and adding the calculation of ship Force.

**Commented [AN26]:** **Comment 14** (R2, general comment): It is not clear why the authors chose two separate locations for the situ and remote sensing study. Is it possible to find satellite image... for the in-situ measurement period?

**Commented [AN27R26]:** Changes:
- A motivation to why the Bornholm study area was chosen for satellite analysis, instead of the Gothenburg study site, has bee... added.
- A comment regarding satellite images for the period of *in situ* measurements at the Gothenburg site has been added.

[revised manuscript text omitted]

**Commented [AN28]:** Comment 9 (R1, specific comment 5): Categorization of the ships in the context of their turbulent wakes does not look physically justified. More reasonable would be to relate the wakes to the ship weight, draught, speed, possibly to the size/number of propellers.

**Commented [AN29R28]:** Changes:
- The wake depth and longevity are now presented in figures relating them to vessel Force, instead of ship type. This provides basis for discussing possible physical explanations to the variation in wake depth and longevity in the detected wakes, as well as the importance of ship size and speed.

[revised manuscript text omitted]

**Commented [AN34]:** **Comment 18** (R2, specific comment): Line 355-359: Why do the bubble wakes look different from turbulence? Please discuss further.

**Commented [AN35R34]:** Changes:
  -The discussion regarding the differences between the bubble a turbulent wake has been expanded.

[Figure]

**Figure 7. Maximum** wake depth for the bubble wake (a, b) and dissipation rate of turbulent kinetic energy (ε) wake (c, d). (a) and (c) are show the close wakes, induced by ships passing at 0–3 ship widths from the instrument, and (b) and (d) show all wakes induced by single ship passages. The x-axis shows the force (F) of the vessel in Newton. Wake depths within the range presented in previous studies are shown in blue and wakes deeper than previously reported are shown in orange.

~~There is no statistically significant correlation between passage distance from the instrument and wake depth, per category or overall. A possible explanation to this lack of correlation could be the skewed data distribution, as there were few passages within 25 m of the instrument. The lack of correlation could also be an indication that the maximum wake depth depends on more variables than just proximity and that further studies are needed to resolve what influences the development of the turbulent wake.Nevertheless, in general the deepest wakes were caused by ships passing closer to the instrument, whereas ships passing at larger distances from the instrument (100–199 m) mainly caused shallower wakes (Fig. 5).Yet, the deepest wakes were caused by ships passing 25–75 m away, which demonstrates that even at distances up to 75 m from the ship, mixing down to 20–30 m depth can be induced (Fig. 5). Even though there is a lack of significant correlation, there are strong indications (Fig. 5) for a distance-dependent detection of maximum wake depths. This in turn indicates that the median wake depths presented in this study could be an underestimation, as it includes wakes from all distance between 0 to 184 m.~~

**3.1.5 Temporal wake longevity**

Figure 8 shows the  wake temporal longevity related to vessel force, for the same wake categories and parameters as in Figure 7,. The median longevity for all wakes was 08:44 min (std 06:29) and 06:30 min (std 03:18) for the bubble and $\varepsilon$  wake respectively) (Table 2). The close wake category had the same longevity for the $\varepsilon$ wake, but the bubble wake was longer at 09:59 min (std 06:34 min). Figure 8 shows no clear correlation between wake longevity and vessel force, for the bubble or $\varepsilon$ wake. Hence, based on the results

> **Commented [AN36]:** **Comment 11** (R1, specific comment 7
> The paper is full of obvious, trivial statements, e.g. "in general th
> deepest wakes were caused by ships passing closer to the instrum
> whereas ships passing at larger distances from the instrument (10
> 199 m) mainly caused shallower wakes : : :" (lines 369-370)

> **Commented [AN37R36]:** Change:
> -With the revised result section, this sentence is removed.

from this study, it seems like parameters related to the vessel speed and size do not explain the variation in wake longevity to a very high degree. However, the relatively low number of passages with a large vessel force makes it difficult to draw any definite conclusions without further studies.

In similarity with the maximum wake depth, the double category had a longer duration on average, compared to the single categories, for both the bubble and $\varepsilon$ wakes (Table 2). A majority of the longest wakes (20–30 min) were induced by ships passing within  3 ship widths of the instrument (Fig. 8).  As this indicates that proximity plays a role in the ability to detect the entire  temporal longevity of the wake, the close wake category median would be a better estimate of wake longevity, compared to   median longevies  calculated from all detected wakesed. Compared with previous studies, a detectable signal of the bubble wake from 10 and up to 30 min, is in agreement (Table 1). Furthermore, the timescale of the wake longevity indicates that in highly trafficked areas, where large ships passes every 10–15 min, there is a high potential of a constant influence of ship-induced vertical mixing

[Figure]

**Figure 8. Wake longevity for the bubble wake (a, b) and dissipation rate of turbulent kinetic energy (ε) wake (c, d). (a) and (c) are show the close wakes, induced by ships passing at 0–3 ship widths from the instrument, and (b) and (d) show all wakes induced by single ship passages. The x-axis shows the force (F) of the vessel in Newton. Wake temporal longevities < 10 min are shown in blue and wake longevities 10–31 min are shown in orange.**

the maximum dissipation rates are in the order of $10^{-4}$ to $10^{-2}$ W kg$^{-1}$ in the core of the wake and decrease with distance from the wake core with a decay length scale of about 70 m. These values are comparable to what one would expect in breaking surface waves, and much larger than what is usually observed in the core of, or below, the surface mixed layer.

[Figure]

Maximum turbulent kinetic energy dissipation rate ($\varepsilon$) intensity [W kg$^{-1}$] in the wake region vs. the passing distance of the ship inducing the wake.

**3.2 Bornholm satellite image analysis**

There was a total of 94 satellite scenes from the period April 2013 to December 2018. Of these scenes, 25 % had a cloud cover of < 23 %, and were analysed for thermal wakes. 48 % of these (n=2311) had visible thermal wakes. The monthly distribution of ship passages and occurrence of thermal wakes are shown in Figure 9. As the number of analysed satellite scenes differed between months, the total number of ship passages for each month was divided by the number of analysed scenes. For all months, the majority of the passages did not induce visible thermal wakes. In April-July, there were several induced thermal wakes per scenes (Fig. 9), most of them in May and June. Occasional thermal wakes were found in September and October, but none were found during the winter months (December–February). In the satellite scenes where thermal wakes were visible, and the environmental conditions were right for thermal wakes to be visible, 21 % of the ship passages induced thermal wakes (Table 3). Looking at all the satellite scenes, including those without environmental conditions appropriate for inducing visible thermal wakes, 10 % of the ship passages induced thermal wakes.

**Commented [AN38]:** Comment 11 (R1, specific comment 7 The paper is full of obvious, trivial statements, e.g. "the maximum dissipation rates : : : in the core of the wake : : :..are : : :..much larger than what is usually observed in the core of, or below, the surface mixed layer" (lines 403-405), etc. etc.

**Commented [AN39R38]:** We acknowledge and understand there are statements in the manuscript that can be perceived as trivial and obvious, depending on the researcher's specialisation. To balance the content to suit a diverse audience, from different highly specialised disciplines, is a general challenge in interdisciplinary research. Therefore, the statement marked here was included for a reason. Our aim is to reach an interdisciplinary audience within ocean science, in line with the scope of this journal. This specific comment was included because the non-oceanographic co-authors of the paper explicitly asked for a comparison between our measured values and values that would occur naturally in the system. We believe that these types of statements fill an important function in making the content of the paper more accessible to an interdisciplinary audience.

However, in the revised result section, this sentence was removed

[Figure]

Figure 9. Seasonal distribution of ship passages for the satellite scenes with < 23 % cloud cover, for the period April 2013 to December 2018. The data labels in the stacked bar indicate the number of passages in each category. As some month has more than one analysed scene, the total number of ship passages for each month was divided by the number of analysed scenes, to get an average number of passages per scene for each month. August had no scenes with < 23 % cloud cover and therefore has no data.

515

~~. The main ship types inducing thermal wakes in the satellite dataset were *Cargo*, *Passenger* and *Tanker*, which all had > 40 passages and constituted 67 % of the total passages. Ship type categories with > 40 passages, but no thermal wakes were sailing (50 passages), pleasure (42 passages), and fishing (83 passages). All other ship types present in the dataset were combined within the *Other* passages category.~~

520

Commented [AN40]: Comment 9 (R1, specific comment 5): Categorization of the ships in the context of their turbulent wakes does not look physically justified. More reasonable would be to re the wakes to the ship weight, draught, speed, possibly to the size/number of propellers.

Commented [AN41R40]: Change:
- The section related to the ship type categories has been remov

[revised manuscript text omitted]

**Commented [AN42]: Comment 19** (R2, specific comment):
Line 550-552: This seems to be a negative result: the stratification
was not affected by the wake. Remove this part. I suggest that the
authors measure more vertical profiles in the study area and/or
provide literature for more data to characterize general and unusual
environmental conditions. 4 casts x 2 days are not enough.

**Commented [AN43R42]:** As mentioned in a previous comment,
we agree and acknowledge that the amount of vertical profiles is too
low to have an in-depth discussion regarding the effects of ship-
induced vertical mixing on stratification. However, for the aim of the
paper, we consider the current amount of profiles enough to describe
the spatiotemporal scales of the turbulent wake.

However, regarding the comment related to the sentence above, we
humbly disagree that it is a negative result. Turbulence will not be
able to reach below the mixed layer, to 17.5 m depth, without mixing
the water. Thus, we find it highly unlikely that the thermal
stratification at 5 m was present within the wake. This means that the
stratification was influenced by the wake and that waters above and
below the thermocline were mixed with each other. However, three
hours later the water has re-stratified and the mixed water has spread
out laterally. We do not claim that there was a long-lasting effect on
the stratification. However, during the longevity of the wake the
stratification was most likely affected. Hence, in a scenario with very
frequent ship passages, there will be less time for the re-stratification
to occur, and a more long-lasting effect on the stratification could be
possible.

For the comment regarding measuring more vertical profiles in the
study area, see **Comment 15.**

**Commented [AN44R42]:** Change:
- The example of the ship passage affecting the thermal
stratification has been clarified.

[revised manuscript text omitted]

**Commented [AN54]:** **Comment 15** (R2, general comment): Also, vertical profiles should have been measured more frequentl see the effect of wakes on stratification and mixing.

**Commented [AN55R54]:** We agree and acknowledge this a main potential improvement of our observations, which we have a discussed in this section of the manuscript. Furthermore, the long term aim of our research is to be able to study and discuss the effe of wakes on stratification and mixing. However, the aim of the current study was to describe the spatiotemporal scales of the turbulent wake, and not resolving all the parameters determining effect of wakes on stratification and mixing. As mention in a prev answer, we realise that the current title and aim, could give the impression that the paper is of a more explanatory nature. Howev we consider the current amount of vertical profiles enough for the aim of this paper and suggest leaving the discussion regarding the effect of wakes on stratification and mixing for the next paper (w high-resolution profiling will be included).

**Commented [AN56R54]:** To address this comment, we will clarify the aim of the paper as being mostly descriptive.

expected to be an important factor for wake depth, it could be one explanation for the absence of statistically significant

760 correlations between wake depth and vessel force other parameters.

**Commented [AN57]:** **Comment 22** (R2, specific comment): Line 618: What parameters?

**Commented [AN58R57]:** Change:
   - The parameters have been specified.

[revised manuscript text omitted]

---

## Author Response (AR2)

**Answers to comments and questions from reviewer 1**

**Comment 1: Line 572 There is undefined concern for wakes leading to suspension of deep sediments at 30m depth. This needs clarification related to the spatial area that would be affected compared to the width of the wake. Then consideration of how large this effect really is compared to tidal or other causes of sediment movement. This could be important for water clarity, organisms sensitive to sediments, etc, but seems unlikely in such a shallow area. Please make more clear.**

We acknowledge that there is a need to clarify that the resuspension observed in this study was due to the surface Kelvin wake, and as the focus of the study is the turbulent wake, the resuspension from the Kelvin wake is considered outside the scope of this paper. To address this comment, the section about resuspension has been clarified and the effects of the ship-induced turbidity has briefly been related to natural phenomena such as swells and waves (line 611-623 in the pdf).

**Comment 2: Figures 4 and 12 show wakes overlaid with the thermal observations of the water surface. Could the authors estimate the total percentage of area that is covered by wakes in the images? What percentage influence due these wakes have compared to the size of the water body? What is the net heat flux or turbulence changes based on the percentage area covered over a day?**

Regarding the total percentage of area covered by the wakes in the image, that number could be calculated. But, as discussed further in the answer to comment 5, there are still uncertainties in the understanding of the coupling between the observed satellite thermal wakes and the vertical distribution of the turbulent wake. However, we acknowledge the interest in giving an estimate of the scale of the impact of ship-induced turbulence. We therefore suggest adding a comparison between the estimated mechanical energy supplied by ship propulsion in the Baltic Sea, and the energy input by winds to the upper surface layer (line: 538-564 in the pdf). We hope that this addition will be satisfactory in contributing an estimation of the scale of the impact.

Moreover, the heat flux is not the main concern here even though we show SST, but the focus is the internal mixing and redistribution of water within the water column. The intense mixing that happens locally and centred in the ship lane region will have a different impact compared with the turbulence changes averaged over the entire area. We are highlighting the extremes and patchiness of the ocean landscape, how anthropogenic activities contributes to this patchiness, and the importance of being aware of these varying conditions within a region for sampling/monitoring and management purposes. We therefore argue that the relative energy input estimation is more adequate to include in this study than an estimate of the heat flux for the entire area.

**Comment 3: Lines 15-16 The authors propose there is a lack of field data on turbulent ship wakes, yet so much research is published on ship wakes, what is the real need?**

We acknowledge that a lot of knowledge exists on turbulent ship wakes under idealized circumstances. However, there is a lack of field data on real turbulent ship wakes in natural stratified water. This is explained in lines 74-88 and in Table 1.

**Comment 4: Line 18. Depth of 32 m ADCP placement in an area with a maximum depth of less than 200 m. This information places ship wakes penetrating to a depth of 18 m.**

We acknowledge that the water depth at the Gothenburg and Bornholm study sites has not been related to the characteristic water depth in the area of interest. To contextualise how well the depth of the study site represent the conditions in the major ship lanes in the Baltic Sea, we have added information about the Baltic Sea and Swedish Westcoast water depth in section 2.1 (line 121-126). The stratification a and water depth has also been mentioned in section 3.3 (line 559-564), to further clarify and contextualise the results. Moreover, as clarified in the answer to comment 6, here the term "deep mixing" is related to previously reported mixing depths and the stratification depth. The term "deep mixing" indicating a mixing deep enough to reach to/across the thermocline, which will be clarified as suggested in the answer to comment 6.

**Comment 5: Line 25. Length of wake is median 13.7 km to 60 km. => maximum of 1.75 hour longevity. Median wake width was 157.5 m without accompanying depth. Usually the wake turbulence rises with the bubbles, leading to a wider, thinner wake influence.**

We acknowledge that we have not included a vertical estimate of the thermal wake. As we did not measure the vertical extent of the thermal wake, we chose not to include it. Nevertheless, we agree that the vertical extent of the thermal wake in the 157.5 m width cross section, probably is not uniform (see further discussion in the next paragraph). One possibility could have been to use the measurements for the turbulent wake to estimate the vertical extent of the thermal wake. We chose not to, as these two methods measure different things: the bubble cloud and turbulent kinetic energy dissipation rate gives a direct indication of the turbulence in the wake and the turbulent wake; the thermal wake is a proxy of the water mass that has been mixed and affected by the ship-induced turbulence and is hence a measure of the wake (but not the turbulence)(as discussed in line 510-519). These measurements give information about different things but are both related to the spatiotemporal extent of ship wakes. However, we will clarify this distinction in the abstract (line 19-21), and it is clarified in the discussion (Line 651-664).

Regarding the shape and vertical extent of the turbulent wake, there are several factors that can affect the "shape" of the wake: environmental conditions such as stratification, currents and waves, as well as vessel related information such as number and rotation direction of the propeller(s), speed and vessel manoeuvring. However, neither the individual nor combined effect of these factors have been well studied, and from the authors experience the one thing we do know is that the shape and extent of the turbulent wake shows large variations. Therefore, we have refrained from general statements or assumptions regarding the vertical distribution of the thermal wake. However, we acknowledge the need for this knowledge, and it is the focus of the follow-up study to this first mapping of the spatiotemporal extent of ship wakes. Still, based on our current knowledge, and the observations made in this study, we suggest addressing this comment by further clarifying the different type of information we get from the thermal and turbulent wake observations, and explain why there is no vertical estimate for the thermal wake.

**Comment 6: Line 552 "deepest wake greater than 15 m" which is listed as "deep vertical mixing" for the Baltic Sea.**

We acknowledge that "deep mixing" is an expression that needs further clarification and contextualisation. In this context, the term "deep" was used in relation to previously reported mixing depths and in relation to the seasonal stratification depth. To address this comment, we have specified the specific depth instead of using the term "deep". To contextualise how this depth relates to the depth and stratification depth in the ship lanes in the Baltic Sea, we have also added information about the general bathymetry and stratification depth. Regarding the comment for Line 552 (now line: 588), what was written was "depths greater than 15 m", and not "deepest wake greater than 15 m". However, there was clearly a need to clarify the meaning of "deep", and we hope the suggested changes addresses the concern in a satisfactory way.

**Grammatical corrections:**
**The word "it" requires an antecedent noun, but is used without in the paper. The writing would be more clear if this was corrected, particularly to stop starting a sentence with the word "it". Lines 352, 469 – the sentences starts with the undefined word "it", while is 15 other occurrences, the words "it is" could be removed and the sentence made more clear. Line 391 – what does "it" mean in this sentence? Line 565 "though it is difficult" What is difficult? Line 351 – Reference Error.**

We would like to thank reviewer 1 for the overall constructive feedback and the suggested grammatical corrections.  The use of the word "it" has been reviewed throughout the paper and exchanged/removed where appropriate. The other grammatical corrections have also been amended.

---

## Author Response (AR3)

June 18th 2021

Dear Editor,

Please find the revised and resubmitted manuscript "*In situ* observations of turbulent ship wakes and their spatiotemporal extent", for consideration for publication in the Ocean Science Special Issue "Shipping and the Environment – From Regional to Global Perspectives". As described by e.g. the two articles by Jalkanen et al. in the special issue, shipping gives rise to chemical, biological and energy pollution. The present manuscript provides another piece to the puzzle of shipping environmental impact studies, introducing characterization of ships' turbulent wakes, which is essential for future understanding of distribution of shipping related chemical and biological pollution from ships and shipping lanes. Regarding energy pollution, while previous studies have focused on ship noise and shoreline erosion/wash, the present manuscript introduces the perspective of ship induced turbulent mixing, supported by observations of mixing down to 30 m depth in the wake.

Studies of ship wakes require an interdisciplinary approach and bridging of the gap between fine-scale hydrodynamics and larger scale physical oceanography. From the thorough review process of the previous two submitted versions of the manuscript, and also feed-back from the scientific community e.g. at the EGU 2021 conference, we have realized that most marine scientists have never considered the potential importance of ship induced turbulence, nor has it been considered by the naval architects specializing in hydrodynamic modeling of ship propulsion. This situation implies that the challenges are not limited to the analytical aspects of bridging spatiotemporal scales used in different disciplines, but the reviews have been strongly influenced by current available knowledge on adjacent topics, which has rather led to a development of the manuscript away from the core scope. For example, requests have been made to include possible induction of sediment resuspension in the manuscript, which then implies that description of the Kelvin wakes needs to be included. This in turn shifts the focus away from the turbulent wake, which is the scope of the article. We believe that this is partly explaining the comment in the last review round, where it is suggested that section 3.3 and 3.4 could be more concisely written. In this revised version, we have therefore aimed at returning to the focus of the scope of the manuscript.

Finally, we recognize that we initially underestimated the complexity of the subject. Yet we believe that this very first manuscript, based on a larger data set than any previous studies on turbulent ship wakes (see table 1 in the revised manuscript), highlights that previous assumptions in literature cannot be verified in data from *in situ* observations. Further, we can show that the duration of the wake signatures calls for consideration e.g. when collecting data by ferrybox setups in shipping lanes; a temperature deviation of up to one centigrade is important to have knowledge about when interpreting ferrybox data.

Detailed responses to the Editor's and Reviewers' comments are provided in the Authors' response to reviewers' comments. All the revisions made can be seen in the uploaded manuscript with tracked changes.

Looking forward to hearing from you.

Yours sincerely,

Amanda Nylund on behalf of all the co-authors

| Nr. | Review comment | Author's answer | Changes in manuscript. All references to row numbers refer to the manuscript with tracked changes included. |
|---|---|---|---|
| 1 | While the authors explain the reasoning behind choice of the two different study regions and two different techniques, there it is still confusion on how the analysis of the Gothenburg Harbour and Bornholm shipping channel are combined to support the findings of the study. Ultimately these two study areas are independent and disconnected from each other. | The aim of this study was to use two different approaches for studies of the spatiotemporal extent of the turbulent wake. The two methods capture different spatiotemporal scales and thus provide complementary information. Using only one of the methods would not be enough, as none of them cover the entire spatiotemporal range of the wake. As stated in previous review rounds, we initially placed instruments at the Bornholm study site, but they were lost. We agree that by making simultaneous *in situ* and *ex situ* observations at the same site, it would be possible to compare and infer between the results from the different methods, which is what we aim to do in future studies. However, in this study we are quantifying different aspects of ship wake dimensions for a large number of ships. For this purpose, we don't consider the different study sites as something that affects the validity/relevance of our conclusions/findings. Rather we use the results of the two approaches as different proxies to study the same phenomenon/process. To address the comment regarding the "*confusion on how the analysis of the Gothenburg Harbour and Bornholm shipping channel are combined to support the findings of the study*", we have made revisions to further clarify how these two different approaches provide different and complementary information regarding the spatiotemporal extent of the turbulent wake. We have also revised the discussion regarding how the results from the two different methodological approaches relate to each other. | Clarifications made with tracked changes in the following sections:
• Abstract (row 15-40)
• Introduction (row 99-103)
• Materials and methods (row 122-136) |

| 2 | The numerous classifications of the data create confusion. For example, the ADCP analysis is divided into "wake", "double", "close wake", "no wake", and "0-3 ship widths", "3-6 ship widths", and "6-55 ship widths". It is unclear why these 6 classifications were used when in the results sections they appear interchangeable i.e., close wake == 0-3 ship widths. Table 3 also has the category of "single" and "all" in addition to "close" and "double". Finally, the statement "Due to the low detection rate in the two larger distance categories, on the close wake category will be used in graphical presentation … (line 351)" suggests that it is only the close wake data that has relevance to supporting the finding of the study. | To address this comment the manuscript has been revised to only include the "wake" and "no wake" division, together with the close wake category (now renamed to "close wake subset" instead and the use of the term categories is no longer used in the manuscript).
The "double wakes" (induced from more than one ship passing the instrument at the same time) has been fully removed from the dataset presented in the result. We agree that the main conclusions regarding wake depth and longevity are based on the close wake subset, but the detection of wakes passing from further distances also provide information about the spatiotemporal extent of the turbulent wake. Therefore, the results from wakes passing from further distances have been moved to the supplementary information. Since there are no publicly available previous studies with such a large dataset of turbulent wake observations, we consider it important to make these results accessible to the scientific community. | Section 2.1.2 Data analysis (row 191-218, 248, 261, 273-279)

Section 3.1 Gothenburg harbour study (row 361-372)

Section 3.1.3 Wake detection rate (row 388-393)

Section 3.1.4 Maximum wake depth (row 406-420, 426-434)

Table 2. (row 423-424)

Fig 7 (row 474)

Section 3.1.5 Temporal wake longevity (row 479-492)

Fig 8 (row 496). |
|---|---|---|---|
| 3 | The supporting figure and table could be improved. For example, Figure 4 caption identifies "ships visible as warmer yellow dots", these are very hard to find. For easy of identification, it would have been appropriate to draw a box around these features. | The figure has been changed to a different figure, in which the ships and wakes are indicated more clearly. The figure also gives a description of the detection and digitalization process. | Figure 4 has been changed (row 345). |
| 4 | X-axis of figures 6,7 and 8 could have been given as F (x107 kg m/s2). | The x-axis title on figure 6, 7, and 8 have been changed to F (x107 kg m/s2). | The x-axis title of figures 6,7 and 8 have been revised. |
| 5 | The sections 3.3 and 3.4 could be more concisely written. | Section 3.3 and 3.4 have been revised and written more concisely. Some of the paragraphs have been moved to other sections of the manuscript and some sections have been removed. Several of the paragraphs/parts of the paragraphs had been added/extended in response to comments from reviewers. It is clear that the topic of the turbulent wake can be viewed from many angles and there are many different aspects that could be discussed in relation to our results. In this revised version of the manuscript we have returned to a clearer focus on the parts we consider within the scope of the current study, which has led to us removing some of the paragraphs/sections written in response to previous reviewer comments. We have tried to find a balance between being more concise and responding to the comments we have received. | See track-changes document for the revised version of section 3.3 and 3.4 (Number of rows reduced from 252 to 167).

Paragraph from original section 3.4 revised and moved to section 3.1.4 (row 445-471) |
| 6 | The conclusion of ship wake impact on the large-scale marine environment is not strongly supported given the local impact of ship wake from both in in-situ (ADCP) and satellite SST study. | We humbly object to this statement, as we have not made any extrapolations/claims of the "ship wake impact on the large-scale marine environment". Based on our results, we conclude that regional (i.e. not large-scale) effects in areas with intense ship traffic cannot be excluded, but further | |

| | | studies are needed to determine when and where these effects are non-negligible. | |
|---|---|---|---|
| 7 | The grammar needs to be improved, as use of the word "it" without an antecedent noun is difficult for a native English speaker to parse. In many other languages, such a vague word is not an issue. In English conversational speaking the word "it" pops up and is understood in context. However, in writing, there are no gestures or social cues as to what this word "it" is referring to. | The use of the word "it" has been revised throughout the manuscript. | See track-changes document. |

---

## Author Response (AR4)

| Nr. | Comment from editor | Author's answer | Changes in manuscript. All references to row numbers refer to the manuscript with tracked changes included. |
|---|---|---|---|
| 1 | I think you might do a little more to address the previous editor's point about integrating the two approaches, and at the same time substantiate ". . turbulent wakes are of a scale relevant to consider when assessing environmental impact from shipping." (end of "Conclusions"). For this, might you integrate a mean ADCP-based ε intensity, duration and vertical extent over a wake width and length corresponding to duration? [Care would be needed regarding width; the thermal wake may be wider than the enhanced-ε wake owing to the relaxation and spreading referred to in line 417.] Such a calculation (relative to ship forcing) might quantify an "amount" of ε related to an "amount" of ship forcing to compare with typical oceanic values of ε. I know this is vague; you are better placed to judge how this might best be done. However, I emphasise ε as the fundamental quantity which those concerned about ocean mixing attempt to measure. | With regards to the part of the comment relating to "substantiating" the statements in the conclusion, we have now rephrased and added some additional lines to further support our argument. See line 674-680.

Regarding the comments about the parameter ε intensity, we agree and acknowledge that ε is an important parameter. However, it is not the mean dissipation rate in the wake, but the integrated dissipation rate at each depth and each vessel that is of interest for mixing. In the present manuscript, we decided that empirical data were not enough to make such an estimate. The aim of this manuscript is to provide a first estimate of the spatiotemporal scales of the turbulent wake and to identify the most relevant parameters and areas to focus on in future studies. We agree that the necessary next step to quantify the turbulent mixing, is to describe the distribution of ε intensity in the wake region, but we consider it a future outlook, rather than part of the present study. . However, we have tried to estimate total energy input from ship traffic and compared with wind energy input between 1 and 20 m in lines 546-578. There, we show that the total integrated dissipation rates from vessels is larger than that from winds during summer months in the Baltic Sea. | Line 674-682. Revised argumentation in conclusion. |
| | **Detailed comments from editor** | | |
| 2 | Line 12. "in" –> "is" | Revised as suggested. | Line 12. "in" revised to "is". |
| 3 | Line 23. "effected" (meaning "caused") or "affected" (meaning "influenced" or "changed")? According to lines 121-122, "affected". | Revised to "affected". | Line 23. "effected" revised to "affected". |
| 4 | Line 114. ADCP and "extent" – is this just vertical or also horizontal extent? | In this study, the ADCP was used to estimate the vertical and temporal extent, which has now been specified. | Line 114. The extent has been specified as "vertical and temporal extent". |
| 5 | Line 192. Omit "that" | Revised as suggested. | Line 192. "That" has been omitted. |
| 6 | Line 198. "origo" –> "the origin"? | Revised as suggested. | Line 198. "origo" has been changed to "the origin". |
| 7 | Lines 213-215, 230-231. Some repetition. | The repeated information has been removed from line 230-231. | Line 230-231. "using the structure function method according to the method described in Lucas et al. (2014)" has been removed. |
| 8 | Line 258. "relates" –> "is used to relate"? | Revised as suggested. | Line 258. "relate" has been revised to "is used to relate". |
| 9 | Line 267. From when until August 2018? Line 446 refers to April 2013 to December 2018 but Figure 9 has none in any August. | Regarding the question about the time period, there was a mistake made in Line 267, it should be "December", not "August". | Line 266-267. "until August 2018" has been removed from line 267 and replaced by "for |

| | | The sentence has been revised to use the term "investigated time period", instead of stating the specific month.

Regarding the comment of the lack of observations in August in figure 9, that is explained in the figure caption. For the entire investigated time period (April 2013 to December 2018) only 23 images had > 23 % cloud cover. None of these images were from the month of August, which is why there is no data for August in figure 9. However, the investigated time span still includes August (for all the years). | the investigated time period" on line 266. |
|---|---|---|---|
| 10 | Line 304. Better ". . local temperature minima (thermal wake centre) and local temperature maxima . ."? | Revised as suggested. | Line 304. "Temperature" added before "minima" and "maxima". |
| 11 | Line 331. "medium" –> "median"? | Revised as suggested. | Line 331. "Medium" changed to "median". |
| 12 | Table 2. Longevity seems to be written as hr:min:sec although stated as [min]. hr: is redundant. This also applies to Supplement table 1. | As suggested, "hr:" has been removed from both tables. | Table 2 and Supplement table 1. The longevity format has been changed from [hr:min:sec] to [min:sec]. |
| 13 | Line 381. "here" –> "there"? | Revised as suggested. | Line 381. "Here" changed to "there". |
| 14 | Line 386. Better ". . wake, the . ." (add comma) | Revised as suggested. | Line 386. A comma was added between "wake" and "the". |
| 15 | Line 391. Better ". . energy, on . ." (add comma) | Revised as suggested. | Line 391. A comma was added between "energy" and "on". |
| 16 | Lines 414-415. I think there is literature about turbulent mixing (efficiency) that could substantiate the "small likelihood", or you might be able to make a comparison as in lines 545-560. What you seem to be saying is that the CTD profiles on retrieval are showing the result of wake-induced mixing and subsequent (during the 3 hours) relaxation as the mixed water (less dense than ambient deeper water) spreads laterally and so becomes shallower. If so, I think this could be made a bit clearer. [If not, then clarification is definitely needed!] | The section (lines 413-425 in track changes manuscript) have been revised to further clarify. The aim with this section is to give an example of an observation where ship-induced turbulence created mixing across the thermocline. We hope that the revised section makes this point in a clearer way.

A reference regarding restratification of intense local mixing has also been added (line 418). | Line 413-425. The sections has been revised for clarity and a reference has been added. |
| 17 | Figure 10B. Omit "median" from horizontal axis label – or are you using a median value for each individual wake? | The values in the figure do represent a median value for the entire wake. As described in the methods section (line 301-303), the wake width was measured in cross profiles along the wake length, in intervals of 250 m. All wakes longer than 750 m therefore had at least three wake width values, from which a median value was calculated. These median values are the ones presented in figure 10B. If needed this could be specified further in the figure caption or result section, but as the method has been explained in the methods section, and the caption is correct, we suggest keeping the current version. | No revisions have been made. |

| 18 | Figure 11. The caption should also explain the other symbols (bars, lines) | An explanation for the box edges and whiskers has been added to the figure caption. | Line 490-494. An explanation of the box and whiskers of the plot has been added. |
|---|---|---|---|
| 19 | Line 506. "dashed" –> "hatched". However, the lines giving hatching in the separation zone are too narrow; not obvious in normal magnification. | As suggested, the word "dashed" has been exchanged to "hatched". The hatching has also been made bolder and larger, to make it visible in normal magnification. | Line 511. "Dashed" has been changed to "hatched". Figure 12 has been revised to have a bolder and larger hatching indicating the separation zone. |
| 20 | Line 667 – data availability. The sentence about "Acoustic measurement data" is not very satisfactory. Please see https://www.ocean-science.net/policies/data_policy.html and "Statement on the availability of underlying data" therein. | The raw data from the ADCP measurements have now been deposited in a FAIR-aligned public data repository and properly cited with a DOI. | Line 684-687. A reference to the deposited dataset has been added. |

---

## Author Response (AR5)

| Nr. | Technical corrections from editor | Author's answer | Changes in manuscript. |
|---|---|---|---|
| 1 | Lines 93-94. ". . wake is needed, to identify . ." (move – or remove – comma) | The comma has been removed. | Line 93. The comma is removed. |
| 2 | Line 146. I don't understand "trough" – please check. | The intended word was through, but the word has now been changed to "over", for clarity. | Line 146. "Trough" has been changed to "over". |
| 3 | Line 218. "Per" –> "By"? | Revised as suggested. (Assuming that it was Line 208, not 218 that was the intended line) | Line 208. "Per" changes to by". |
| 4 | Lines 348-349. "Due to . . widths." – incomplete sentence (redundant?) | The incomplete sentence has been removed. | Line 348-349. The indicated sentence has been removed. |
| 5 | Lines 431, 432. "Temporal" is probably unnecessary. Better order is "Wake temporal longevity". | Revised as suggested. | Line 431. The heading has been revised to "Wake temporal longevity |
| 6 | Section 3.2.1 including heading, and line 662. "longevity" tends to imply time and "extent" or "length" might be better for the spatial aspect. | We acknowledge this comment and have revised the usage of "spatial longevity" in the manuscript, changing it to "length". | Line 26. "spatial longevity" is changed to "length". Line 318, 461, 469, 662. "longevity" changed to "length". |
| 7 | Line 483. "adaption" –> "adjustment"? | Revised as suggested. | Line 483. "Adaption" changed to "adjustment". |
| 8 | Lines 545-546. 3.9 GW is a rate (I guess average through the year) and not a "yearly input". | We appreciate this comment, as the sentence is indeed confusing. We have now changed "yearly" to "total", as that was the intended meaning. The word "yearly" has been added in relation to the distance travelled by each ship type, as those were the yearly values used in the calculation. We hope this clarifies the sentence. | Line 545. "yearly" has been added before "…distance travelled by…" and the previous "yearly" has been changed to "total". |
| 9 | Line 596. "form" –> "from" | Revised as suggested. | Line 596. "form" has been changed to "from". |
| 10 | Line 601. "depth" (typo) | Revised as suggested. | Line 601. "Dept" changed to "depth". |
| 11 | Line 624. Omit "that" | Revised as suggested. | Line 624. "That" has been omitted. |
| 12 | Line 666. Better "depth. Wind mixing is homogeneously distributed over the Baltic Sea, whereas vessel mixing . ." | Revised as suggested. | Line 666. "While" has been removed at the beginning of the sentence and exchanged to "whereas" after the comma. |